evolution, genomics

ecological speciation, adaptive radiation, novelty, speciation genomics, adaptive introgression, trophic innovation

**Author for correspondence:**
Christopher H. Martin
e-mail: chmartin@berkeley.edu

# We get by with a little help from our friends: shared adaptive variation provides a bridge to novel ecological specialists during adaptive radiation

Emilie J. Richards[1,2] and Christopher H. Martin[1,2]

[1]Department of Integrative Biology and [2]Museum of Vertebrate Zoology, University of California, Berkeley, CA, USA

CHM, 0000-0001-7989-9124

Adaptive radiations involve astounding bursts of phenotypic, ecological and species diversity. However, the microevolutionary processes that underlie the origins of these bursts are still poorly understood. We report the discovery of an intermediate *C.* sp. 'wide-mouth' scale-eating ecomorph in a sympatric radiation of *Cyprinodon* pupfishes, illuminating the transition from a widespread algae-eating generalist to a novel microendemic scale-eating specialist. We first show that this ecomorph occurs in sympatry with generalist *C. variegatus* and scale-eating specialist *C. desquamator* on San Salvador Island, Bahamas, but is genetically differentiated, morphologically distinct and often consumes scales. We then compared the timing of selective sweeps on shared and unique adaptive variants in trophic specialists to characterize their adaptive walk. Shared adaptive regions swept first in both the specialist *desquamator* and the intermediate 'wide-mouth' ecomorph, followed by unique sweeps of introgressed variation in 'wide-mouth' and de novo variation in *desquamator*. The two scale-eating populations additionally shared 9% of their hard selective sweeps with the molluscivore *C. brontotheroides*, despite no single common ancestor among specialists. Our work provides a new microevolutionary framework for investigating how major ecological transitions occur and illustrates how both shared and unique genetic variation can provide a bridge for multiple species to access novel ecological niches.

## 1. Introduction

Rapid bursts of diversification and repeated bouts of speciation like those seen in adaptive radiations contradict current mechanistic speciation models that predict diversification should slow with time as available niche space becomes increasingly subdivided and disruptive selection becomes weaker with each recurrent speciation event (e.g. [1–3]). Diversification on complex adaptive landscapes with multiple empty fitness peaks corresponding to different niches provides an alternative mechanism to niche subdivision [4–6]. However, these landscapes present a new problem to our mechanistic understanding of adaptive radiations: how do populations manage to escape local optima, cross fitness valleys and access new fitness peaks [7–10]? Colonizing new fitness peaks on the adaptive landscape presents challenges because it requires transitions in behaviours, morphological traits or a combination of the two that allow organisms to adapt to new ecological niches [11]. Spectacular ecological transitions do often occur during adaptive radiation, such as blood-drinking [12] or plant carnivory [13,14], yet it is still poorly understood how such seemingly discontinuous transitions occur.

Recent conceptual frameworks for understanding adaptation to novel fitness peaks suggest that these major ecological transitions probably occur in stages of potentiation, actualization and refinement [15,16]. The initial emergence of a

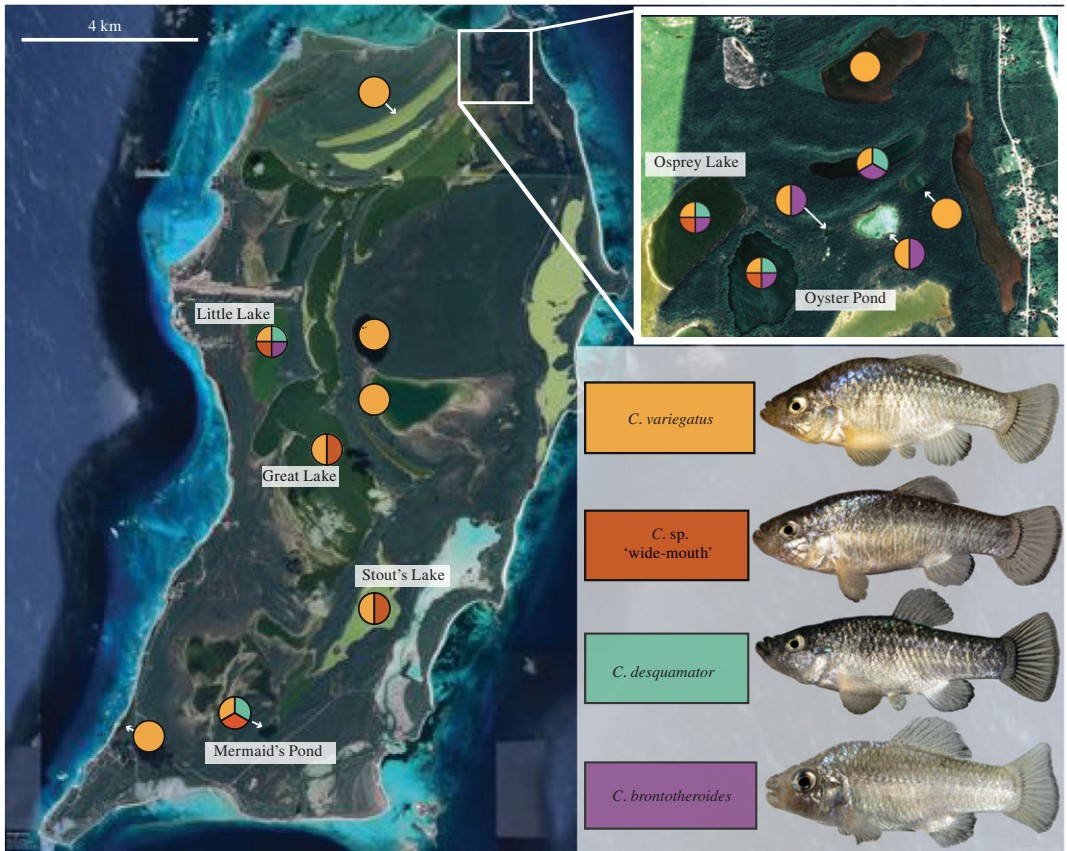

**Figure 1.** The SSI radiation of pupfish. Pie charts indicate the presence of sympatric *Cyprinodon* species in each lake and are colour coded with representative pictures of generalist *C. variegatus* (gold), recently discovered *C. sp.* 'wide-mouth' ecomorph (red-orange) with intermediate jaws, scale-eater *C. desquamator* (teal) with the largest oral jaws, and molluscivore *C. brontotheroides* (purple) with characteristic nasal protrusion. Labelled lakes contain all known *Cyprinodon* populations on the island that were sampled for this study. Satellite image from Google Earth. (Online version in colour.)

novel trait probably requires further refinement to become successfully incorporated into the functional ecology of an organism. Several experimental laboratory studies suggest that novel ecological transitions are highly contingent on accruing a series of mutations that incrementally refine adaptations to colonize new fitness peaks [16,17]. This idea that genetic background is important in setting the stage for adaptation also underlies many hypotheses for adaptive radiation, such as the hybrid swarm and syngameon hypotheses—in which radiations are driven by acquiring novel combinations of alleles through the exchange of genetic variation either from distinct lineages outside the radiation or within the radiation itself [18]. However, we are only just beginning to explore how gene flow and shared genetic variation give recipient lineages access to new fitness peaks in the wild and generate adaptive radiations [6].

An adaptive radiation of trophic specialist pupfishes on San Salvador Island (SSI) in the Bahamas is an excellent system for understanding how the rapid evolution of major ecological transitions occurs in nature. This radiation contains a widespread generalist pupfish species (*Cyprinodon variegatus*) that occurs in sympatry with two previously described trophic specialists that are endemic to the hypersaline lakes on the island: a molluscivore (*C. bronotheroides*) with a novel nasal protrusion which is an oral-sheller of gastropods [19] and a scale-eating specialist (*C. desquamator*) with two-fold larger oral jaws [20]. The evolutionary novelties in this system originated recently; the lakes on SSI were dry during the last glacial maximum 6–20 kya years ago [21,22]. Intriguingly, we recently discovered a fourth species of pupfish living in sympatry with the two specialists and generalist

on SSI [23]. This species exhibits intermediate jaw morphology between *C. desquamator* and *C. variegatus* (figure 1). Here, we refer to this new ecomorph as *C. sp.* 'wide-mouth' because its mouth is wider than any other species in the radiation. The multi-peak fitness landscape driving this radiation suggests that *C. desquamator* is isolated by a large fitness valley from *C. variegatus* and *C. brontotheroides* [9], and this intermediate *C. sp.* 'wide-mouth' may provide clues about the microevolutionary processes underlying how the observed novel fitness peaks are traversed in the wild.

Here, we first investigated the position of *C. sp.* 'wide-mouth' on the ecological spectrum from generalist to scale-eating specialist using a combination of morphological, behavioural, dietary and genomic data. We then estimated the demographic history of the 'wide-mouth' and explored the spatial origins and timing of selection on shared and unique genetic variation involved in adaptation to scale-eating to better understand this ecological transition. Our results suggest that while intermediate in jaw length, which is known to be functionally relevant for the highly specialized scale-eater *C. desquamator*, *C. sp.* 'wide-mouth' demonstrates transgressive morphology and a distinct genetic background. Our investigation of the timing of selection and genetic origins of the adaptive alleles shared and unique between the two scale-eating species indicates divergent adaptive walks that were highly dependent on their distinct genetic backgrounds. Despite shared origins, access to unique genetic variation in each of the two scale-eating sister species probably resulted in distinct adaptive walks and ultimately contributed to the diversity of ecological specialists observed in this radiation.

## 2. Methods

### (a) Ecological and morphological characterization of 'wide-mouth' scale-eater

*C. variegatus*, *C. desquamator* and *C.* sp. 'wide-mouth' individuals from three lake populations (Osprey Lake, Great Lake and Oyster Pond) in which we had sufficient specimens (n = 84; *C. brontotheroides* not shown) were measured for nine external morphological traits using digital calipers. Traits were selected for specific connections to foraging performance which differed across the three species in a previous study [9]. We also characterized diet for *C. variegatus*, *C. desquamator* and *C.* sp. 'wide-mouth' in Osprey Lake from stomach content analyses (n = 10 per species) and stable isotope analyses of muscle tissue from wild-collected samples (n = 75). Dietary overlap was characterized by comparing population mean scale count from gut contents using ANOVA, ellipse areas and bivariate means on isotope biplots using SIBER [24]. See electronic supplementary material for more details on sample sizes and analyses.

### (b) Genomic library preparation and variant filtration

To explore the evolutionary history of *C.* sp. 'wide-mouth', we sequenced whole genomes of 22 individuals following protocols used in a previous study [25] that included genomes from *C. variegatus*, *C. desquamator* and *C. brontotheroides*. Our final genetic dataset after filtering contained 6.4 million variants across 110 individuals from the four species (7.9 x median coverage). See electronic supplementary material for the full sequencing and genotyping protocol.

### (c) Genomic origins of the 'wide-mouth' scale-eater

We first tested whether these *C.* sp. 'wide-mouth' individuals represented recent (e.g. F1/F2) hybrids of *C. variegatus* and *C. desquamator* in the wild using principal component and ADMIXTURE analyses to look for the genome-wide pattern expected in PCAs when recent hybrids between two populations are included. We also used formal tests for introgression and admixed populations, $f_3$ and $f_4$-statistics [26], to assess whether *C.* sp. 'wide-mouth' are the byproduct of recent admixture. Finally, we used *fastsimcoal2* (v. 2.6.0.3) [27], a demographic modelling approach based on the folded minor allele frequency spectrum (mSFS), to discriminate among alternative evolutionary scenarios for the origin of 'wide-mouth' and estimated divergence times among all four species based on the best model fit from AIC (see electronic supplementary material for more detail).

### (d) Characterization of unique and shared adaptive alleles among specialists

Across all four populations in Osprey Lake, we looked for regions that showed evidence of a hard selective sweep using SweeD (v. 3.3.4) [28]. The composite likelihood ratio (CLR) for a hard selective sweep was calculated in 50 kb windows across scaffolds that were at least 100 kb in length (99 scaffolds; 85.6% of the genome). Significance thresholds were determined using CLR values from neutral sequences simulated under MSMC inferred demographic scenarios of historical effective population size changes (electronic supplementary material, figure S1 and table S1).

Next, we searched for candidate adaptive alleles associated with species divergence by overlapping selective sweep regions with regions of high genetic divergence based on fixed or nearly fixed SNPs between species. We chose to also look at regions with nearly fixed SNPs ($F_{st} \geq 0.95$) to accommodate ongoing gene flow among these young species. $F_{st}$ between the populations and species was calculated per variant site using the weir-pop-fst function in vcftools (v. 0.1.15) [29].

### (e) Timing of selection on candidate adaptive alleles

We also determined the relative age of candidate adaptive alleles by generating estimates of coalescent times using starTMRCA (v. 0.6.1) [30]. For each candidate adaptive allele that was unique to the three specialists and the 16 shared alleles between *C. desquamator* and *C.* sp. 'wide-mouth', a 1 Mb window surrounding the variant was extracted into separate vcfs for each species. These sets of variants were then analysed in starTMRCA with a mutation rate of $1.56 \times 10^{-8}$ substitutions per base pair (estimated from Caribbean pupfishes [25]) and a recombination rate of $3.11 \times 10^{-8}$ (from stickleback [31]). Each analysis was run three times per focal adaptive allele, and all runs were checked for convergence between and within runs. Most runs rapidly converged within the initial 6000 steps, but five runs did not converge after an additional 4000 steps and were discarded from further analysis. See electronic supplementary material for more details on timing analyses.

### (f) Characterization of adaptive introgression in 'wide-mouth' scale-eater

Lastly, we investigated the spatial origins of adaptive alleles shared and unique to the two scale-eating specialists by searching in our previous study spanning Caribbean-wide outgroup populations for these alleles [25]. Adaptive alleles were assigned as standing genetic variation if observed in any population outside SSI or de novo if they were only observed within populations on SSI. Additionally, we investigated signatures of introgression across the genome of *C.* sp. 'wide-mouth' and *C. desquamator* to determine if they showed evidence of adaptive introgression from outgroup generalist populations as observed previously [25]. See methods section of the electronic supplementary material for more details on introgression analyses.

## 3. Results

### (a) 'Wide-mouth' scale-eater is ecologically intermediate and morphologically distinct

We found that the 'wide-mouth' ecomorph is morphologically distinct from *C. desquamator* and *C. variegatus* across a suite of craniofacial traits (figure 2a,b). The lower jaw length of *C.* sp. 'wide-mouth' was intermediate between *C. desquamator* and *C. variegatus* (figure 2c), while the mouth width and adductor mandibulae height were 8% larger in *C.* sp. 'wide-mouth' than *C. desquamator* (figure 2d,e). These morphological differences were consistent across multiple lakes (electronic supplementary material, figure S2). Small modifications in craniofacial morphology among these species have major impacts on scale-eating performance in this system by altering kinematic traits such as peak gap size (which is partially controlled by the length of the lower jaw), jaw protrusion distance and the angle of the lower jaw relative to the suspensorium [32].

*C.* sp. 'wide-mouth' also did not show morphological divergence comparable to that observed in the molluscivore *C. brontotheroides*. The molluscivore specialist presents an opposing axis of morphological divergence to the scale-eating specialists, with shorter oral jaw length and larger eye diameter than even the generalist *C. variegatus*, in

*Proc. R. Soc. B* **289**: 20220613

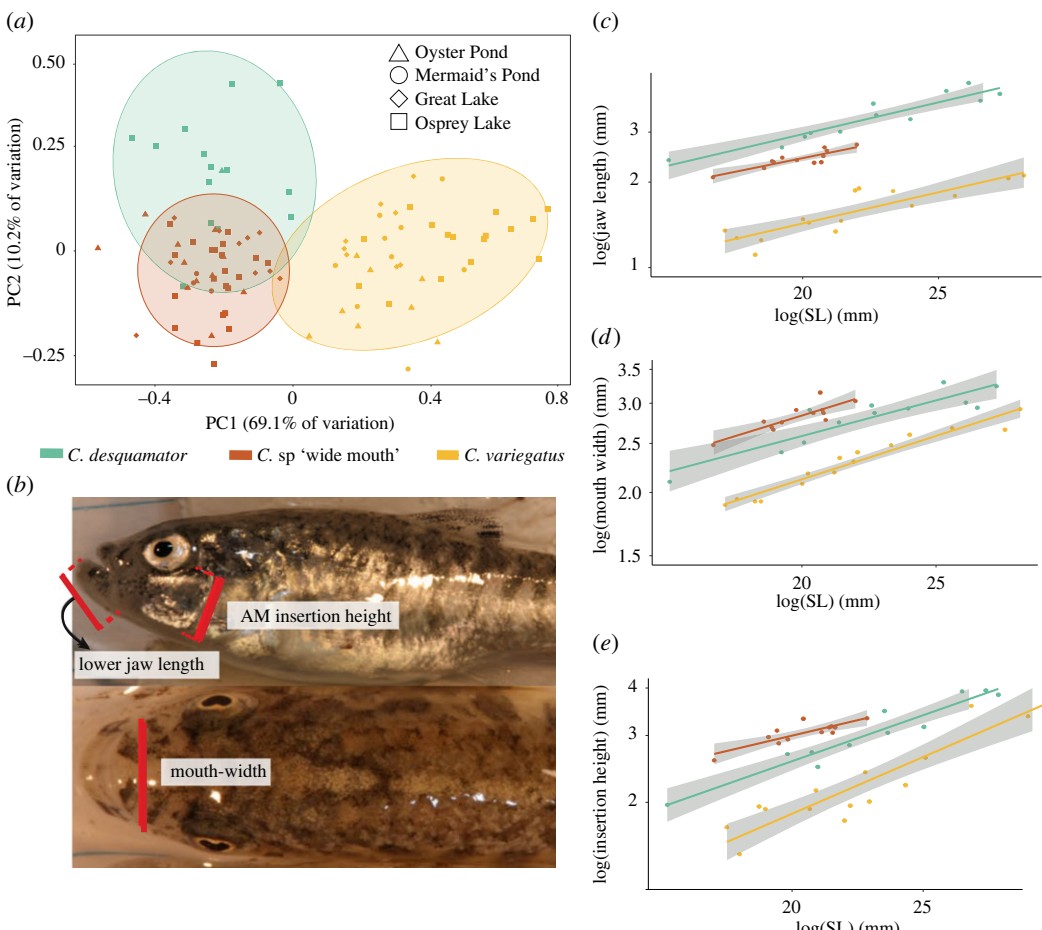

**Figure 2.** *C.* sp. 'wide-mouth' has distinct morphology within the SSI adaptive radiation. (*a*) First two principal components of morphological diversity for eight size-corrected traits and 95% confidence ellipses by species (*C. variegatus*: gold; *C.* sp. 'wide-mouth': red-orange; *C. desquamator*: teal; *C. brontotheriodes* not shown). PC1 is mainly described by lower jaw length and PC2 by adductor mandibulae insertion height, mouth width and neurocranium width. (*b*) Depictions of the three external measurements that best distinguished *C.* sp. 'wide-mouth' from both *C. desquamator* and *C. variegatus*, measured using digital calipers. (*c*–*e*) The relationship between standard length (mm) of individuals and their (*c*) lower jaw length, (*d*) buccal cavity width and (*e*) adductor mandibulae insertion height (AM insertion) across individuals of the three species in Osprey Lake. 95% confidence bands for linear models in grey. (Online version in colour.)

addition to a novel nasal protrusion of the maxilla not observed in any other Cyprinodontidae species [33].

Morphological traits were heritable in a common garden laboratory environment after one generation: laboratory-reared *C.* sp. 'wide-mouth' displayed significantly larger mouth width than *C. desquamator* (*t*-test; $p = 0.003$) and maintained their characteristic intermediate jaw lengths (ANOVA; $p = 0.03$, electronic supplementary material, figure S3). There was also some evidence of phenotypic plasticity in both laboratory-reared *C. desquamator* and *C.* sp. 'wide-mouth' compared to wild individuals likely caused by the common laboratory diet. See electronic supplementary material for more details.

### (b) 'Wide-mouth' scale-eater occupies a distinct intermediate scale-eating niche

We found that *C.* sp. 'wide-mouth' ingested scales, but at a significantly lower frequency than *C. desquamator* (Wilcoxon Rank Sum test, $p = 0.004$; figure 3*a*). We did not detect any scales in *C. variegatus* guts (figure 3*a*). Detritus made up the rest of the *C.* sp. 'wide-mouth' and *C. desquamator* diets and was the dominant component of *C. variegatus* gut contents, except for a single individual with one mollusk shell. A previous study that characterized contents of *C. variegatus*, *C. brontotheroides* and *C. desquamator* populations across several ponds also

found detritus to be the dominant component of each species' diet (49–71%) and nearly zero scales in the gut contents of *C. variegatus* and *C. brontotheroides* [33].

The intermediate scale-eating dietary niche of *C.* sp. 'wide-mouth' was complemented by our stable isotope analyses, which provide long-term snapshots of the carbon sources and relative trophic position in these species. Osprey Lake individuals collected on the same day from the same site differed in δ15N levels across species (ANOVA, $P = 4.55 \times 10^{-6}$; figure 3*b* and electronic supplementary material, figure S4); *C.* sp. 'wide-mouth' δ15N was intermediate between *C. variegatus* and *C. desquamator* (Tukey HSD; $P = 1.34 \times 10^{-5}$ & $1.11 \times 10^{-4}$ respectively), supporting its intermediate trophic position. Additionally, SIBER analyses indicated distinct trophic positioning based on the lack of extensive overlap in niche space measured by standard ellipse areas and bivariate means with 95% confidence intervals of isotope values among the species (figure 3*b*).

### (c) *C.* sp. 'wide-mouth' did not result from hybridization between *Cyprinodon variegatus* and *Cyprinodon desquamator*

Several lines of genomic evidence from PCA, ADMIXTURE, and *f-statistic* analyses support the *C.* sp. 'wide-mouth'

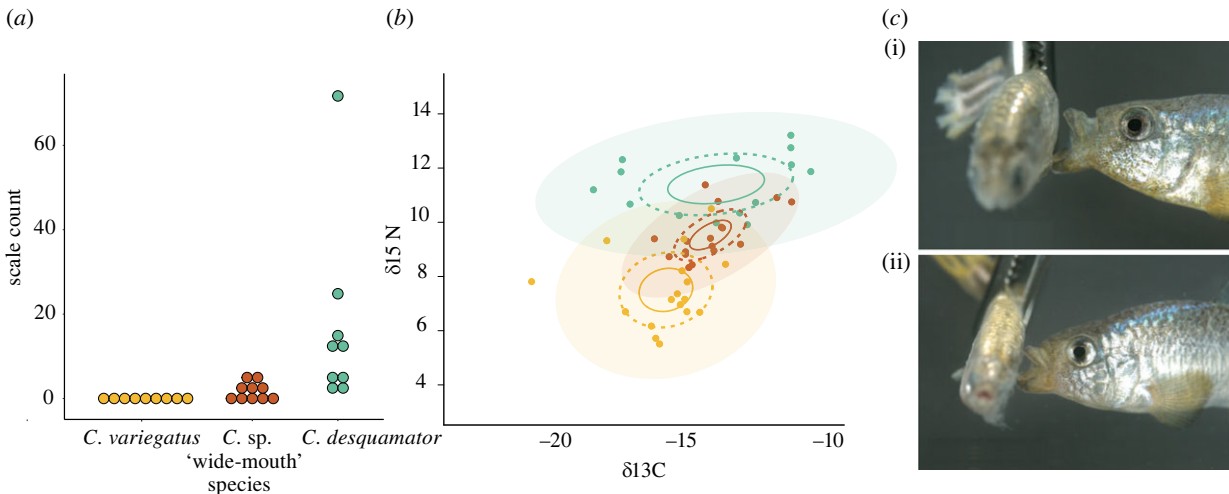

**Figure 3.** *C.* sp. 'wide-mouth' ingests scales. (*a*) Scale counts from gut content analysis of the hindgut of Osprey Lake pupfish populations (10 individuals per species). (*b*) Relative trophic position (δ15 N stable isotope ratio) and dietary carbon source (δ13C stable isotope ratio) with 95% confidence ellipses for generalist and scale-eating species. Solid lines represent 95% confidence intervals around bivariate mean, dotted lines represent standard ellipse areas corrected for sample size (contain 40% of data; SEAc), shaded circles represent ellipse area that contain 95% of the data calculated using the R package SIBER. (*c*) Still images of scale-eating strikes in (i) *C. desquamator* and (ii) *C.* sp. 'wide-mouth' filmed at 1100 fps on a Phantom VEO 440S camera. (Online version in colour.)

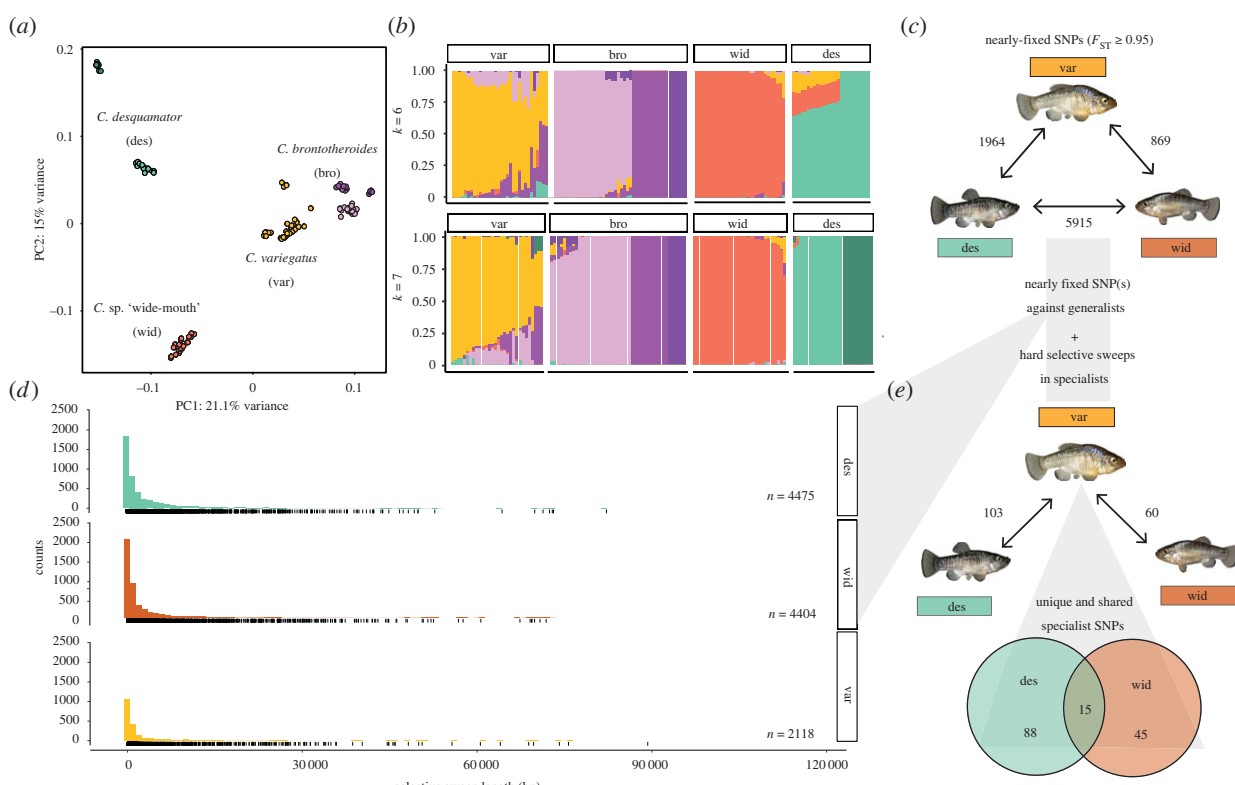

**Figure 4.** Patterns of selection and genetic divergence in specialist genomes. (*a*) Principal components analysis of the four focal groups on SSI based on an LD-pruned subset of genetic variants (78 840 SNPs). (*b*) Ancestry proportions across individuals of the four focal groups. Proportions were inferred from ADMIXTURE analyses with two values of *K* with the highest likelihood on the same LD-pruned dataset in (*a*). (*c*) The total number of fixed or nearly fixed SNPs ($F_{ST} \geq 0.95$) between each group in Osprey Lake. (*d*) Selective sweep length distributions across generalist and scale-eating species. Rug plot below each histogram represents the counts of selective sweeps in different length bins. (*e*) The number of adaptive alleles (fixed or nearly fixed SNPS [$F_{ST} \geq 0.95$]) relative to *C. variegatus* and under selection in the scale-eating specialists in Osprey Lake. Venn diagram highlights those adaptive alleles that are unique to each specialist and shared with the other specialist. (Online version in colour.)

ecomorph as a genetically distinct species rather than a recent hybrid between *C. desquamator* and *C. variegatus* on SSI (figure 4*a–c*; electronic supplementary material, figures S5 and S6 for more details). Demographic modelling of divergence and gene flow on SSI places *C.* sp. 'wide-mouth' as

sister to *C. desquamator,* supporting previous phylogenetic inference [23]. In the best-supported model of 28 demographic models tested (electronic supplementary material, table S2), 'wide-mouth' and *C. desquamator* diverged 11 658 years ago (95 CI: 8257–20 113 years; electronic supplementary material,

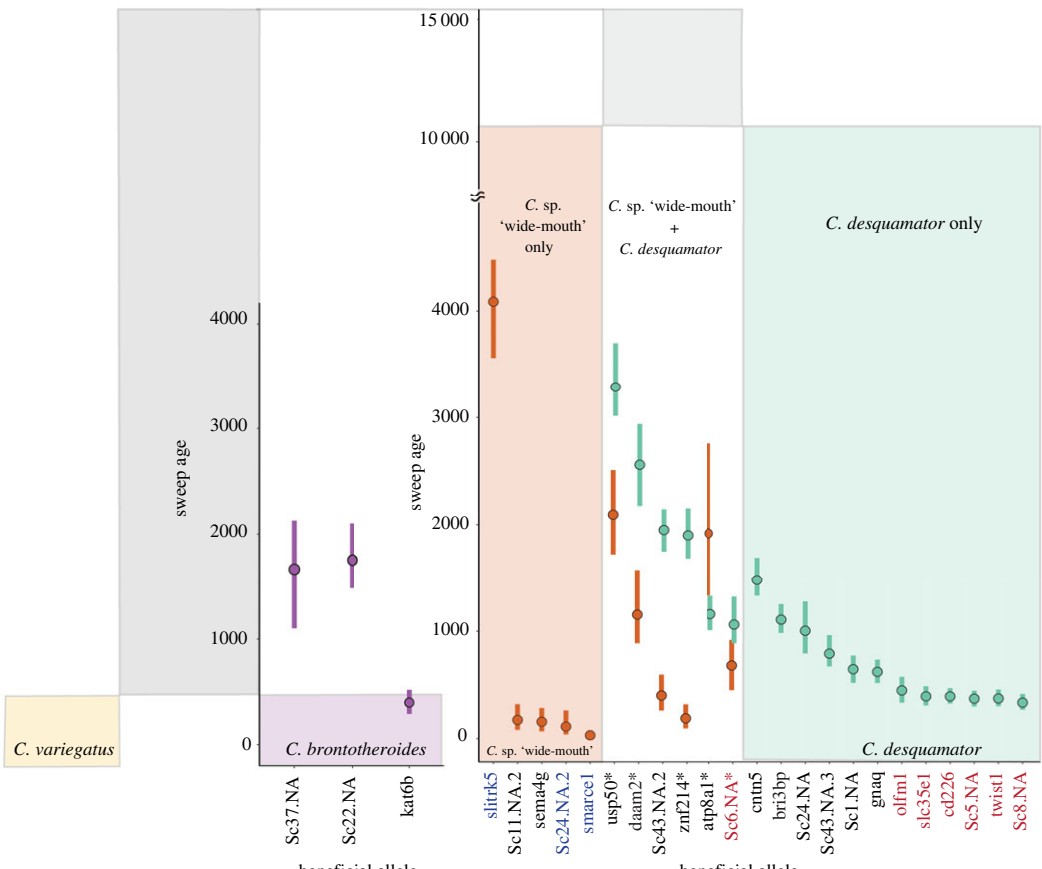

**Figure 5.** Timing of selection on adaptive alleles in trophic specialists nested within the demographic history of the radiation. The median and 95% HPD estimates for the timing of selection on sets of fixed or nearly fixed SNPs (named by the gene they are in or within 20 kb of) for the three specialist populations found in sympatry in Osprey Lake (sweeps in *C. variegatus* not shown). The age of each beneficial allele is colour coded by the species and the inferred demographic history is displayed in the background for comparison. Gene names highlighted in bold are associated with oral jaw size. Gene names are coloured by source of genetic variation (de novo: red; introgressed with outgroup: blue; standing genetic variation: black). Gene names with asterisk indicate those inferred as introgressed between *C. desquamator* and *C.* sp. 'wide-mouth'. (Online version in colour.)

figure S7 and table S2) with ongoing gene flow. Additionally, *C. desquamator* and *C.* sp. 'wide-mouth' were more genetically diverged from each other than to the generalist *C. variegatus* (e.g. $F_{st}$ in figure 4c).

### (d) C. sp. 'wide-mouth' scale-eater contains both shared and unique adaptive alleles

Next we looked at regions of the genome in both *C. desquamator* and *C.* sp. 'wide-mouth' that showed strong evidence of hard selective sweeps. We found six shared hard selective sweeps in both species containing a total of 15 SNPS that were fixed or nearly fixed compared to the sympatric generalist *C. variegatus* (figure 4e): 10 SNPs were in unannotated regions, two were in the introns of the gene *daam2*, and three were in putative regulatory regions (with 20 kb) of the genes *usp50*, *atp8a1* and *znf214* (one variant each). Shared adaptive alleles in the gene *daam2*, a wnt signalling regulator, are intriguing because knockdown of this gene causes abnormal snout morphology in mice [34] and abnormal cranial and skeletal development in zebrafish [35].

We also found unique sets of adaptive alleles in *C.* sp. 'wide-mouth' and *C. desquamator* (figure 4e). None of the adaptive alleles unique to *C.* sp. 'wide-mouth' were in or near genes annotated for craniofacial phenotypes in model organisms, despite its distinctive craniofacial morphologies.

In *C. desquamator,* three of 12 unique adaptive alleles were in or near genes associated with or known to affect craniofacial phenotypes: a *de novo* non-synonymous coding substitution in the gene *twist1*, several putative regulatory variants near the gene *gnaq* and eight variants in or near the gene *bri3 bp*, which is located inside a QTL region for cranial height in pupfish [36]. *C. brontotheriodes* also contained at least one unique candidate craniofacial adaptive allele: a non-synonymous coding substitution in the gene *kat6b* (figure 5), which is associated with abnormal craniofacial morphologies, including shorter mandibles, in mice [37]. This pattern of unique alleles relevant to craniofacial phenotypes in specialists *C. brontotheriodes* and *C. desquamator*, but not *C.* sp. 'wide-mouth', holds even if we lower the threshold to the top 1 percentile of $F_{ST}$ outliers between specialists and generalist (see electronic supplementary material, results; figures S9 and S10).

### (e) The origins of adaptive alleles in C. sp. 'wide-mouth' and desquamator scale-eaters

The adaptive alleles shared by *C. desquamator* and *C.* sp. 'wide-mouth' occurred as low frequency standing genetic variation in the Caribbean, with the exception of a single de novo allele on SSI located in an unannotated region on scaffold 6 (figure 5; electronic supplementary material,

table S3). The adaptive alleles unique to *C. desquamator* and *C.* sp. 'wide-mouth' also predominantly came from standing genetic variation (84% and 81%, respectively). Fourteen per cent of adaptive alleles unique to *C. desquamator* were de novo mutations on SSI and 2% occurred in candidate introgression regions (electronic supplementary material, table S4). We found the opposite in *C.* sp. 'wide-mouth': only 4% of their unique adaptive alleles were de novo mutations whereas 15% occurred in candidate introgression regions (electronic supplementary material, table S4). This adaptive introgression occurred with generalist populations sampled from North Carolina and Laguna Bavaro in the Dominican Republic (electronic supplementary material, table S5 and figure S11). Using the Relative Node Depth (RND) statistic, we also discovered that five of the six shared adaptive alleles (all except for the unannotated region on scaffold 43; electronic supplementary material, table S6) appear introgressed between *C. desquamator* and *C.* sp. 'wide-mouth', suggesting a substantial contribution of introgression to the adaptive alleles observed in scale-eating specialists.

## (f) Timing of selection on adaptive alleles reveals features of the adaptive walk to scale-eating

Selective sweeps occurred much more recently in both populations than their inferred divergence times (figure 5). Intriguingly, selection on four of the six adaptive alleles occurred significantly earlier in *C. desquamator* than *C.* sp. 'wide-mouth'. Only a single adaptive allele had an older median age estimate in *C.* sp. 'wide-mouth' than *C. desquamator*, although the 95% HPD intervals overlapped between the species (figure 5). Additionally, overall we found a significant difference in timing of selection between shared and unique adaptive alleles in the two scale-eater populations (ANOVA *p*-value = 0.00478). In *C. desquamator*, shared adaptive alleles swept before any unique adaptive alleles (Tukey HSD *p*-value = 0.003217; figure 5). In *C.* sp. 'wide-mouth', shared adaptive alleles with *C. desquamator* also generally swept before those unique to the species, despite unique alleles originating from standing and introgressed variation (figure 5). However, this difference in timing between shared and unique adaptive alleles in *C.* sp 'wide-mouth' was not significant due to one unique adaptive allele (*slitrk5*) that swept first (figure 5; ANOVA, Tukey HSD; *p* = 0.8367). This adaptive allele resides in a region that appears to be introgressed with the Laguna Bavaro generalist population in the Dominican Republic where this allele also show signs of a hard selective sweep [25]. The older age estimate of this sweep in *C.* sp 'wide-mouth' might be due to older shared selection for the alleles in other Caribbean populations before introgression with *C.* sp 'wide-mouth'. All other introgressed adaptive alleles in *C.* sp. 'wide-mouth' swept more recently than shared sweeps with *C. desquamator*, including the shared de novo allele, and were not under selection in outgroup generalist populations.

Intriguingly, all but one of the de novo adaptive alleles in *C. desquamator* swept at the same time in the recent past (figure 5). Only one of these adaptive alleles in *olfm1* region had a 95% HPD age range that overlapped with the next oldest selective sweep of standing genetic variation (*gnaq*; figure 5), suggesting a discrete stage of selection on de novo mutations in *C. desquamator*.

## (g) Shared signatures of selection across the three specialists in the radiation

Lastly, we compared the genetic divergence and selection patterns observed in the two scale-eating specialists to the divergent molluscivore specialist *C. brontotheroides* to investigate the extent of allele sharing among all three trophic specialists in this adaptive radiation. We found that no fixed or nearly fixed alleles relative to the generalist *C. variegatus* were shared across all three specialists (electronic supplementary material, figures S9 and S10; electronic supplementary material, results). However, we did find evidence of 44 shared selective sweeps across all three specialist populations that were not shared with *C. variegatus* populations (electronic supplementary material, figure S12). These shared regions were significantly enriched for genes annotated for metabolic processes (electronic supplementary material, figure S12), suggesting shared selection for metabolizing the more protein-rich diet across all three trophic specialists (also see [38]).

# 4. Discussion

## (a) Discovery of a new cryptic intermediate scale-eater highlights the power of reusing genetic variation to access novel niches

The hallmark of adaptive radiation is a rapid burst of diversification which is predicted by theory to slow down over time as niche subdivision increases [6]. An alternative possibility is that radiations can be self-propagating and that the diversity generated within the first stages of radiation helps beget further diversity [39]. This could happen through exploitation of new trophic levels created by new species or physical alterations of the environment by new species that may create additional opportunities for speciation (reviewed in [6,40]). The 'diversity begets diversity' hypothesis can also be visualized as the exploration of a complex multi-peaked fitness landscape; as species in the radiation colonize new peaks, this provides access to additional neighbouring fitness peaks to fuel rapid radiation. Our discovery of a cryptic new scale-eating species through morphological, dietary and genomic analyses revealed shared nearly fixed or fixed adaptive alleles in both scale-eating species relative to the generalist *C. variegatus*. While *C.* sp. 'wide-mouth' is ecologically intermediate in its scale-eating behaviour, our estimates of the relative timing of selective sweeps suggest that these shared alleles were first selected upon in the more specialized scale-eater *C. desquamator*, although unaccounted for demographic differences may also be contributing to differences in estimated timing between species.

Intriguingly, the shared adaptive alleles between *C. desquamator* and *C.* sp. 'wide-mouth' have potentially introgressed recently rather than selected upon in their shared common ancestor. Five of the six regions surrounding these shared adaptive alleles showed patterns of high genetic similarity consistent with introgression (electronic supplementary material, table S6). Alternatively, this genetic similarity may also be caused by strong background selection on shared ancestral variation. Effective population sizes are not drastically different between the two species and exon density is not in the upper tail of the genome wide-distribution (electronic supplementary material, figure S1

and table S6), two conditions in which background selection tends to confound adaptive introgression inferences [41,42]. However, we do not have extensive knowledge of recombination breakpoints in this non-model system to distinguish between strong background selection on shared ancestral variation and adaptive introgression scenarios for each candidate adaptive introgression region.

We also found strong signatures of introgression in *C.* sp. 'wide-mouth' genomes from outgroup generalist populations that were not shared with *C. desquamator* (electronic supplementary material, figure S11 and table S7). Craniofacial morphology is a major axis of diversification between trophic specialists in this system [43], yet *C.* sp. 'wide-mouth' appears to have little unique genetic variation relevant for craniofacial traits compared to the other two specialists (electronic supplementary material, figure S10). Despite this, they do exhibit transgressive craniofacial phenotypes not seen in the other specialists. An intriguing implication of these findings is that hybridization may allow different species to share many of the same adaptive alleles to occupy distinct but similar niches, in line with the syngameon and 'diversity begets diversity' hypotheses of adaptive radiation [18,39].

## (b) An adaptive walk underlies the major ecological transition from generalist to scale-eating specialist

The foundational model of adaptation is that it proceeds in 'adaptive walks' towards fitness optima that involve the sequential fixation of adaptive alleles that move a population in the phenotypic direction of the local optimum [44]. The distinct timing of selection across different adaptive alleles in both *C. desquamator* and *C.* sp. 'wide-mouth' suggests that the ecological transition from generalist to novel scale-eating specialist involved an adaptive walk in which selection on a beneficial allele was contingent on prior fixation of other adaptive alleles in each specialists' genetic background (see electronic supplementary material for further discussion). This is best highlighted by the pattern observed in *C. desquamator* in which nearly all de novo mutations swept at the same time in a distinct selective stage from other adaptive variants rather than being selected upon as they originated (figure 5).

## (c) The (un)predictability of adaptive walks to novel ecological niches

A recent study characterizing genotypic fitness landscapes underlying the transition from *C. variegatus* and *C. desquamator* revealed a rugged landscape with many local fitness peaks, likely due to epistatic interactions among alleles [45]. The staggered timing of selection on alleles lends support to this finding. Epistasis can reduce the number of adaptive walks selection will promote [46] and might explain why the same adaptive alleles were the first to undergo hard selective sweeps in both *C.* sp. 'wide-mouth' and *C. desquamator*.

We also found evidence for shared selective sweeps across all three specialists in regions that were enriched for genes annotated for metabolic processes such as short-chain fatty acid and propionate metabolism (electronic supplementary material, figure S12D). The lack of fixed alleles in these regions relevant to dietary specialization suggests polygenic selection (see electronic supplementary material for more discussion). Subtle shifts of allele frequencies across the genome

can lead to divergent genomic backgrounds that give populations access to different ecological niches (e.g. [47,48]).

While both shared sweeps among all specialists and shared adaptive alleles among the two scale-eating species suggest constrained adaptive walks along overlapping genotypic pathways, we still see most selective sweeps are unique to each species in this radiation (figure 4; electronic supplementary material, figure S11). Curiously, some adaptive standing genetic variation rose to high frequency in *C. desquamator* but did not similarly undergo selection in *C.* sp. 'wide-mouth', despite its adaptation to a similar ecological niche and the presence of these alleles segregating at low frequency in *C.* sp. 'wide-mouth' populations. This highlights the dual influence of epistatic interactions on adaptive walks in rugged landscapes—epistasis reduces number of available paths but increases the number of local fitness peaks populations can get stuck on [49]. Selection on the same adaptive alleles may have allowed both scale-eating species access to the same area of the fitness landscape but epistatic interactions with private mutations and introgressed variation in each lineage may have resulted in divergent paths to scale-eating, ultimately contributing to diverse evolutionary outcomes even from a shared starting point.

The use of adaptive alleles from distinct spatial sources, the distinct morphologies and divergent genomic backgrounds, and potential introgression of adaptive alleles from the more specialized scale-eater *C. desquamator* into *C.* sp. 'wide-mouth' reveals a tangled path for novel ecological transitions in nature. The complex epistatic interactions at microevolutionary scales implicated in this study make it all the more fascinating that novel ecological transitions are a common macroevolutionary feature of adaptive radiation.

**Ethics.** Pupfishes were euthanized in an overdose of buffered MS-222 (Finquel, Inc.) following approved protocols from the University of North Carolina at Chapel Hill Animal Care and Use Committee (no. 18-061.0), and the University of California, Berkeley Animal Care and Use Committee (no. AUP-2015-01-7053) and preserved in 95–100% ethanol. All animals were collected and exported with 2017 and 2018 permits from the Bahamas Environmental Science and Technology commission and Ministry of Agriculture.

**Data accessibility.** All genomic data are archived at the National Center for Biotechnology Information BioProject Database (Accessions: PRJNA833158, PRJNA690558; PRJNA394148, PRJNA391309 and PRJNA305422). Genomic analyses, ecological and morphological data are available through the Dryad Digital Repository: https://doi.org/10.6078/D18H9B [50]. Scripts are available at https://github.com/emiliejri chards/Cyprinodon_sp_wide_mouth_ecomorph.

The data are provided in the electronic supplementary material [51].

**Authors' contributions.** E.J.R.: conceptualization, data curation, formal analysis, funding acquisition, investigation, methodology, resources, validation, visualization, writing—original draft and writing—review and editing; C.H.M.: conceptualization, data curation, funding acquisition, investigation, methodology, project administration, resources, supervision, validation and writing—review and editing.

Both authors gave final approval for publication and agreed to be held accountable for the work performed therein.

**Conflict of interest declaration.** The authors declare no competing interests.

**Funding.** This research was funded by the National Science Foundation DEB CAREER grant no. 1749764, National Institutes of Health grant no. 5R01DE027052-02, Binational Science Foundation grant 2016136, the University of North Carolina at Chapel Hill, and the University of California, Berkeley to C.H.M.

**Acknowledgements.** We thank Priya Moorjani, Michelle St. John, Joseph McGirr, Jacquelyn Galvez, David Tian, Austin Patton, Joseph Heras and three anonymous reviewers for helpful comments on the manuscript; the Gerace Research Centre and Troy Day for logistical support; and the government of the Bahamas for permission to collect and export samples.

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
