## [Peer Review File · Proceedings of the Royal Society B: Biological Sciences]

Review History

RSPB-2021-1782.R0 (Original submission)

Review form: Reviewer 1

Recommendation

Accept with minor revision (please list in comments)

Scientific importance: Is the manuscript an original and important contribution to its field?

Excellent

General interest: Is the paper of sufficient general interest?

Excellent

Quality of the paper: Is the overall quality of the paper suitable?

Excellent

Is the length of the paper justified?

Yes

Should the paper be seen by a specialist statistical reviewer?

No

Do you have any concerns about statistical analyses in this paper? If so, please specify them explicitly in your report.

No

It is a condition of publication that authors make their supporting data, code and materials available - either as supplementary material or hosted in an external repository. Please rate, if applicable, the supporting data on the following criteria.

Is it accessible?

Yes

Is it clear?

Yes

Is it adequate?

Yes

Do you have any ethical concerns with this paper?

No

Comments to the Author

In this ms, Richards & Martin describe a new 'intermediate' wide-mouth scale-eater ecomorph in the pupfish adaptive radiation in the Bahamas using morphological and dietary analyses, and use whole genome resequencing data to investigate their evolution. They find that the new wide-mouth ecomorph is morphologically and trophically intermediate between the generalist and specialised scale-eater. Using genomic data, they find that the two scale-eater ecomorphs have diverged over 10kya, and that, contrary to expectations, that the shared 'scale-eater'-adaptive alleles swept earlier in the specialised scale-eater *D. desquamator* compared to the intermediate wide-mouth ecomorph. Comparisons of absolute divergence across the genome indicate that this pattern is potentially due to introgression of adaptive alleles between scale-eater ecomorphs.

Overall, this is a very interesting study, and I liked the approach of using this ecomorphologically intermediate ecomorph to test hypotheses on the evolution of the scale-eater species and the underlying adaptive alleles. I have very much enjoyed reading this paper and only have a few comments (see below).

L. 185 – 191 (L. 219 – 227 in Suppl.): While the morphological differences are quite pronounced in the wild caught individuals, they seem to be quite a lot weaker in the lab-reared once. I was wondering if the differences are statistically significant, as 2 – 4% seem quite small and the confidence intervals overlap quite a bit for insertion height (Fig. S3A). Also, is there any indication of how biologically relevant such small morphological differences are in this system? I am not suggesting that they are not meaningful, I am more intrigued about any information, which would be interesting to include in the discussion (especially since morphological differences are not discussed in the main discussion).

L. 259: Do you mean fig. 5E here?

L. 287: You refer to the GO term panel here. I think there went something wrong with the figure panel order. But figure 5 and the legend are very nice and clear otherwise.

L. 289 – 290: How do you know that these SNPs are in regulatory regions of those genes? Were those SNPs annotated using a particular software, e.g. snpEFF or did you define regulatory regions based on the distance to the gene? Would be good to provide this information.

L. 350 – 359: While low dxy values are a good indicator for the introgression of these alleles from one specialist into the other, I was wondering if you have considered using more specialised approaches to test for the presence of introgression around shared adaptive loci compared to the remainder of the genome (e.g. RNDmin; Rosenzweig et al. 2016)? This would give the argument more weight and support the conclusions regarding the shared adaptive alleles a bit more.

Review form: Reviewer 2

Recommendation

Major revision is needed (please make suggestions in comments)

Scientific importance: Is the manuscript an original and important contribution to its field?

Acceptable

General interest: Is the paper of sufficient general interest?

Good

Quality of the paper: Is the overall quality of the paper suitable?

Good

Is the length of the paper justified?

Yes

Should the paper be seen by a specialist statistical reviewer?

No

Do you have any concerns about statistical analyses in this paper? If so, please specify them explicitly in your report.

No

It is a condition of publication that authors make their supporting data, code and materials available - either as supplementary material or hosted in an external repository. Please rate, if applicable, the supporting data on the following criteria.

Is it accessible?

Yes

Is it clear?

Yes

Is it adequate?

Yes

Do you have any ethical concerns with this paper?

No

Comments to the Author

The manuscript by Richards et al. presents 24 genomes of a new weekly-specialized scale eater pupfish species that occurs sympatrically with three other species, including a strongly specialized scale eater to which the new wide-mouth species is sister. The manuscript combines a diversity of well-chosen analyses to generate a good understanding of the radiation and particularly of the wide-mouth species which had previously not been studied genomically.

Major points:

While I do agree that diversity can beget diversity, I do not see how this manuscript adds evidence to this hypothesis. Having this even in the title is thus not correct in my opinion. I would recommend strongly toning down this part.

In the manuscript, the authors imply independent evolution of scale eating aided by introgression. However, the two scale eating species show up as sister species in all analyses. The finding of most sharing of «specialist alleles» between *desquamator* and *wide-mouth* is to be expected as these «specialist alleles» are inferred as alleles divergent from *variegatus*. As *desquamator* and *wide-mouth* are sister species, all alleles fixed in their common ancestor would be included here and would be shared by both scale eating species even if they are not involved in adaptation to scale eating. The fact that *C. brontotheriodes* had fewest alleles is also expected, given that it is very closely related to *variegatus*.

Similarly, the particularly low D_{xy} between the two scale-eating species at genomic regions with shared adaptive alleles does not imply introgression. It could just be a region of strong background selection or less gene flow with other species. As the two scale-eating species are sister species, no introgression is needed for allele sharing. The authors argue that this is unlikely because the age of the sweep is inferred to be different between the species. However, all those regions with low D_{xy} show the smallest difference in the selective sweep age, whereas the two regions with average or high D_{xy} show the largest difference in sweep age between the two species. The order of sweep age in the low D_{xy} regions is broadly consistent between the two species, indicating that they are correlated. I do not know the caveats of this method well enough to judge it, but I assume some demographic differences could contribute to the consistent shift in sweep age between the species. In almost all cases, the sweep age of shared alleles is older than that of unique alleles. This again suggests that the shared alleles could just be ancestral alleles, whereas the unique alleles are more recent alleles. None of the alleles that swept very recently are shared, speaking against recent introgression.

The authors have previously shown that introgression from San Salvador Island facilitated the evolution of the specialized scale eater. It would be good to better link this manuscript to previous work by showing if the unique alleles and the shared alleles are also found in the San Salvador Island population. In Table S7, the authors show this for the shared alleles. However, I would also like to see it for the unique alleles. Are the unique specialist alleles newly arisen or do they also confer old alleles that perhaps introgressed from other geographic regions? Given that only half of the *C. sp.* 'wide-mouth' individuals contained scales in the stomach content. What was the rest of their diet? This would be very important to know.

Minor points:

L. 58: Unclear what stages means here. The two cited papers do not talk about stages...

Why is the molluscivore (*C. brontotheriodes*) not included in the phenotypical and stomach content analysis? This seems relevant.

Table S1: Including an admixture test testing for a topology that is inconsistent with the genome-wide average tree and the demographic models, is more confusing than helpful in my opinion. I would remove the bold lines.

Fig. 6 is beautiful and a very nice summary of all the ages.

Decision letter (RSPB-2021-1782.R0)

01-Oct-2021

Dear Dr Martin:

I am writing to inform you that your manuscript RSPB-2021-1782 entitled "We get by with a little help from our friends: diversity begets diversity through shared adaptive genetic variation" has, in its current form, been rejected for publication in Proceedings B.

This action has been taken on the advice of referees, who have recommended that substantial revisions are necessary. With this in mind we would be happy to consider a resubmission, provided the comments of the referees are fully addressed. However please note that this is not a provisional acceptance.

Sincerely,
Professor Gary Carvalho
<mailto:proceedingsb@royalsociety.org>

Associate Editor
Comments to Author:

This study examining microevolutionary mechanisms underlying an adaptive radiation has now been reviewed by two experts in the field. The reviews were mixed. Referee 1 was positive about the importance and general interest for readers of PRSB, while Referee 2 was more critical and thought the paper was only marginally relevant and that the conclusions about adaptive introgression were overstated and not supported by the data. Specifically, both Referees highlighted that possibility that alternative processes other than introgression may explain the low Dxy between the two scale-eating species at genomic regions with shared adaptive alleles, to the extent that some of the data could be interpreted as counter to the introgression hypothesis (Referee 2). Referee 1 also points out that additional approaches to test for the presence of introgression around shared adaptive loci compared to the remainder of the genome would have been beneficial to support these conclusions. I also agree with both Referees on the need for some

ecological data to support assumptions about these species. On balance, both Referees suggest that it may be possible to reconcile these deficiencies with additional data and analyses that could significantly improve the evidence for introgression of adaptive alleles and increase the potential interest of the study to readers of PRSB . Consequently, I am recommending that resubmission be allowed.

Reviewer(s)' Comments to Author:

Referee: 1

Comments to the Author(s)

In this ms, Richards & Martin describe a new 'intermediate' wide-mouth scale-eater ecomorph in the pupfish adaptive radiation in the Bahamas using morphological and dietary analyses, and use whole genome resequencing data to investigate their evolution. They find that the new wide-mouth ecomorph is morphologically and trophically intermediate between the generalist and specialised scale-eater. Using genomic data, they find that the two scale-eater ecomorphs have diverged over 10kya, and that, contrary to expectations, that the shared 'scale-eater'-adaptive alleles swept earlier in the specialised scale-eater *D. desquamator* compared to the intermediate wide-mouth ecomorph. Comparisons of absolute divergence across the genome indicate that this pattern is potentially due to introgression of adaptive alleles between scale-eater ecomorphs.

Overall, this is a very interesting study, and I liked the approach of using this ecomorphologically intermediate ecomorph to test hypotheses on the evolution of the scale-eater species and the underlying adaptive alleles. I have very much enjoyed reading this paper and only have a few comments (see below).

L. 185 – 191 (L. 219 – 227 in Suppl.): While the morphological differences are quite pronounced in the wild caught individuals, they seem to be quite a lot weaker in the lab-reared once. I was wondering if the differences are statistically significant, as 2 – 4% seem quite small and the confidence intervals overlap quite a bit for insertion height (Fig. S3A). Also, is there any indication of how biologically relevant such small morphological differences are in this system? I am not suggesting that they are not meaningful, I am more intrigued about any information, which would be interesting to include in the discussion (especially since morphological differences are not discussed in the main discussion).

L. 259: Do you mean fig. 5E here?

L. 287: You refer to the GO term panel here. I think there went something wrong with the figure panel order. But figure 5 and the legend are very nice and clear otherwise.

L. 289 – 290: How do you know that these SNPs are in regulatory regions of those genes? Were those SNPs annotated using a particular software, e.g. snpEFF or did you define regulatory regions based on the distance to the gene? Would be good to provide this information.

L. 350 – 359: While low dxy values are a good indicator for the introgression of these alleles from one specialist into the other, I was wondering if you have considered using more specialised approaches to test for the presence of introgression around shared adaptive loci compared to the remainder of the genome (e.g. RNDmin; Rosenzweig et al. 2016)? This would give the argument more weight and support the conclusions regarding the shared adaptive alleles a bit more.

Referee: 2

Comments to the Author(s)

The manuscript by Richards et al. presents 24 genomes of a new weekly-specialized scale eater pupfish species that occurs sympatrically with three other species, including a strongly specialized scale eater to which the new wide-mouth species is sister. The manuscript combines a diversity of well-chosen analyses to generate a good understanding of the radiation and particularly of the wide-mouth species which had previously not been studied genomically.

Major points:

While I do agree that diversity can beget diversity, I do not see how this manuscript adds evidence to this hypothesis. Having this even in the title is thus not correct in my opinion. I would recommend strongly toning down this part.

In the manuscript, the authors imply independent evolution of scale eating aided by introgression. However, the two scale eating species show up as sister species in all analyses. The finding of most sharing of «specialist alleles» between *desquamator* and wide-mouth is to be expected as these «specialist alleles» are inferred as alleles divergent from *variegatus*. As *desquamator* and wide-mouth are sister species, all alleles fixed in their common ancestor would be included here and would be shared by both scale eating species even if they are not involved in adaptation to scale eating. The fact that *C. brontotheriodes* had fewest alleles is also expected, given that it is very closely related to *variegatus*.

Similarly, the particularly low D_{xy} between the two scale-eating species at genomic regions with shared adaptive alleles does not imply introgression. It could just be a region of strong background selection or less gene flow with other species. As the two scale-eating species are sister species, no introgression is needed for allele sharing. The authors argue that this is unlikely because the age of the sweep is inferred to be different between the species. However, all those regions with low D_{xy} show the smallest difference in the selective sweep age, whereas the two regions with average or high D_{xy} show the largest difference in sweep age between the two species. The order of sweep age in the low D_{xy} regions is broadly consistent between the two species, indicating that they are correlated. I do not know the caveats of this method well enough to judge it, but I assume some demographic differences could contribute to the consistent shift in sweep age between the species. In almost all cases, the sweep age of shared alleles is older than that of unique alleles. This again suggests that the shared alleles could just be ancestral alleles, whereas the unique alleles are more recent alleles. None of the alleles that swept very recently are shared, speaking against recent introgression.

The authors have previously shown that introgression from San Salvador Island facilitated the evolution of the specialized scale eater. It would be good to better link this manuscript to previous work by showing if the unique alleles and the shared alleles are also found in the San Salvador Island population. In Table S7, the authors show this for the shared alleles. However, I would also like to see it for the unique alleles. Are the unique specialist alleles newly arisen or do they also confer old alleles that perhaps introgressed from other geographic regions? Given that only half of the *C. sp.* 'wide-mouth' individuals contained scales in the stomach content. What was the rest of their diet? This would be very important to know.

Minor points:

L. 58: Unclear what stages means here. The two cited papers do not talk about stages...

Why is the molluscivore (*C. brontotheriodes*) not included in the phenotypical and stomach content analysis? This seems relevant.

Table S1: Including an admixture test testing for a topology that is inconsistent with the genome-wide average tree and the demographic models, is more confusing than helpful in my opinion. I would remove the bold lines.

Fig. 6 is beautiful and a very nice summary of all the ages.

Author's Response to Decision Letter for (RSPB-2021-1782.R0)

See Appendix A.

RSPB-2022-0613.R0

Review form: Reviewer 1

Recommendation

Accept with minor revision (please list in comments)

Scientific importance: Is the manuscript an original and important contribution to its field?

Good

General interest: Is the paper of sufficient general interest?

Good

Quality of the paper: Is the overall quality of the paper suitable?

Excellent

Is the length of the paper justified?

Yes

Should the paper be seen by a specialist statistical reviewer?

No

Do you have any concerns about statistical analyses in this paper? If so, please specify them explicitly in your report.

No

It is a condition of publication that authors make their supporting data, code and materials available - either as supplementary material or hosted in an external repository. Please rate, if applicable, the supporting data on the following criteria.

Is it accessible?

Yes

Is it clear?

Yes

Is it adequate?

Yes

Do you have any ethical concerns with this paper?

No

Comments to the Author

The authors have addressed all my comments, and the addition of new analyses, results and discussion sections has greatly improved the manuscript.

Decision letter (RSPB-2022-0613.R0)

27-Apr-2022

Dear Dr Martin

I am pleased to inform you that your Review manuscript RSPB-2022-0613 entitled "We get by with a little help from our friends: shared adaptive variation provides a bridge to novel ecological specialists during adaptive radiation" has been accepted for publication in Proceedings B.

The referee(s) do not recommend any further changes. Therefore, please proof-read your manuscript carefully and upload your final files for publication. Because the schedule for publication is very tight, it is a condition of publication that you submit the revised version of your manuscript within 7 days. If you do not think you will be able to meet this date please let me know immediately.

To upload your manuscript, log into <http://mc.manuscriptcentral.com/prsb> and enter your Author Centre, where you will find your manuscript title listed under "Manuscripts with Decisions." Under "Actions," click on "Create a Revision." Your manuscript number has been appended to denote a revision.

You will be unable to make your revisions on the originally submitted version of the manuscript. Instead, upload a new version through your Author Centre.

- 1) A text file of the manuscript (doc, txt, rtf or tex), including the references, tables (including captions) and figure captions. Please remove any tracked changes from the text before submission. PDF files are not an accepted format for the "Main Document".
- 2) A separate electronic file of each figure (tiff, EPS or print-quality PDF preferred). The format should be produced directly from original creation package, or original software format. Please note that PowerPoint files are not accepted.
- 3) Electronic supplementary material: this should be contained in a separate file from the main text and the file name should contain the author's name and journal name, e.g. `authurname_procb_ESM_figures.pdf`

All supplementary materials accompanying an accepted article will be treated as in their final form. They will be published alongside the paper on the journal website and posted on the online figshare repository. Files on figshare will be made available approximately one week before the accompanying article so that the supplementary material can be attributed a unique DOI. Please see: <https://royalsociety.org/journals/authors/author-guidelines/>

4) Data-Sharing and data citation

It is a condition of publication that data supporting your paper are made available either in the electronic supplementary material. Authors must complete the 'data accessibility' section in the submission system. This should list the database and accession number for all data from the article that has been made publicly available, for instance:

NB. From April 1 2013, peer reviewed articles based on research funded wholly or partly by RCUK must include, if applicable, a statement on how the underlying research materials – such as data, samples or models – can be accessed.

[http://datadryad.org/submit?journalID=RSPB&manu=\(Document not available\)](http://datadryad.org/submit?journalID=RSPB&manu=(Document%20not%20available)) which will take you to your unique entry in the Dryad repository. If you have already submitted your data to dryad you can make any necessary revisions to your dataset by following the above link.

Please include the **Dryad DOI in the Data Accessibility section** and reference in the paper's bibliography.

Please see our Data Sharing Policies (<https://royalsociety.org/journals/authors/author-guidelines/>).

Once again, thank you for submitting your manuscript to *Proceedings B* and I look forward to receiving your final version. If you have any questions at all, please do not hesitate to get in touch.

Sincerely,
Professor Gary Carvalho
<mailto:proceedingsb@royalsociety.org>

Associate Editor

Comments to Author:

The authors have done an excellent job of addressing the concerns raised in the previous submission. I agree with the Referee that the additional introgression analyses significantly bolster their support for the ecological intermediacy of this new species, which will be of significant broad interest to readers of PRSB as well as advancing the mechanisms contributing to adaptive radiation. For all of these reasons, I am pleased to recommend acceptance of the MS.

Reviewer(s)' Comments to Author:

Referee: 1

Comments to the Author(s).

The authors have addressed all my comments, and the addition of new analyses, results and discussion sections has greatly improved the manuscript.

Decision letter (RSPB-2022-0613.R1)

03-May-2022

Dear Dr Martin

I am pleased to inform you that your manuscript entitled "We get by with a little help from our friends: shared adaptive variation provides a bridge to novel ecological specialists during adaptive radiation" has been accepted for publication in *Proceedings B*.

Data Accessibility section

Open Access

Paper charges

Sincerely,

Proceedings B

Appendix A

UNIVERSITY OF CALIFORNIA, BERKELEY

BERKELEY · DAVIS · IRVINE · LOS ANGELES · MERCED · RIVERSIDE · SAN DIEGO · SAN FRANCISCO

BARBARA · SANTA CRUZ

Department of Integrative Biology
Museum of Vertebrate Zoology
University of California, Berkeley
Berkeley, CA 94720-3140 U.S.A.

919-971-8633
chmartin@berkeley.edu

March 30th, 2022

Dear Dr. Gary Carvalho,

Please find the attached revised manuscript “**We get by with a little help from our friends: shared adaptive variation provides a bridge to novel ecological specialists during adaptive radiation**” that we hope you will consider for publication in *Proceedings of the Royal Society B*.

The following modifications have been made:

- 1) We addressed the helpful comments and concerns of the reviewers by extensively revising the entire manuscript, including clarifying methods, adding new statistical analyses of introgression and spatial genetic variation, revising discussion of the underlying mechanisms, adding interpretations of the patterns observed, revising the title and focus of the manuscript, and revising Figures 3-5 to reflect these changes.
- 2) We added more discussion and statistical analyses of the differences in ecology across species in this radiation, which is largely based on their dietary differences. We placed the dietary results in the context of previous studies that characterize the dietary niches of pupfish species San Salvador Island and more broadly across generalist species that span the Caribbean range of *Cyprinodon*. We include statistical analyses to assess how distinct the ecological niche of this new cryptic intermediate scale-eating species is based on its diet.
- 3) We added several formal tests of introgression to address our questions about the evolutionary scenario underlying the shared adaptive alleles between the two scale-eating species. The 4 regions inferred to potentially be introgressed between the two species were additionally supported as candidate introgressed regions from more formal RND test of introgression. We also detected an additional region to be a candidate introgressed region under this new test, bring the total to 5 of 6 regions surrounding shared adaptive alleles exhibiting signatures of introgression.
- 4) We added additional characterization of the spatial origins of adaptive alleles that are unique to each scale-eater species to complement the previously characterized spatial origins of only the shared adaptive alleles. This additional characterization led to an intriguing new finding of introgression from a distant outgroup generalist population in

North Carolina into the *C. sp* 'wide-mouth' genomes. We followed this finding of introgression with additional analyses to characterize the history of admixture involving this group, where we discovered a hybrid origin for *C. sp* 'wide-mouth' from admixture between the more specialized scale-eater *C. desquamator* and distantly related generalist outgroup population. This new finding helped us address reviewer concerns about the independent evolution of scale-eating aided by introgression (see point 5).

- 5) We toned down our original wording to avoid implying independent evolution of scale-eating aided by introgression and strong evidence of diversity begetting diversity by removing it from the title and revising much of the main text to incorporate more caveats. The focus of the manuscript has shifted based on new results spurred by the suggestions and questions from the reviewers.
- 6) We added additional discussion of the new findings, 2 figures and 4 tables to the supplement to further address the reviewers' comments.

Please do not hesitate to contact us if you have any questions or concerns.

Sincerely,

Emilie Richards

PhD candidate

Department of Integrative Biology
University of California, Berkeley
3101 Valley Life Sciences Building
Berkeley, CA 94720-3160
ejrichards@berkeley.edu

Christopher Martin

Assistant Professor

Department of Integrative Biology
University of California, Berkeley
3186 Valley Life Sciences Building
Berkeley, CA 94720-3160
chmartin@berkeley.edu

Associate Editor
Comments to Author:

This study examining microevolutionary mechanisms underlying an adaptive radiation has now been reviewed by two experts in the field. The reviews were mixed. Referee 1 was positive about the importance and general interest for readers of PRSB, while Referee 2 was more critical and thought the paper was only marginally relevant and that the conclusions about adaptive introgression were overstated and not supported by the data. Specifically, both Referees highlighted that possibility that alternative processes other than introgression may explain the low Dxy between the two scale-eating species at genomic regions with shared adaptive alleles, to the extent that some of the data could be interpreted as counter to the introgression hypothesis (Referee 2). Referee 1 also points out that additional approaches to test for the presence of introgression around shared adaptive loci compared to the remainder of the genome would have been beneficial to support these conclusions. I also agree with both Referees on the need for some ecological data to support assumptions about these species. On balance, both Referees suggest that it may be possible to reconcile these deficiencies with additional data and analyses that could significantly improve the evidence for introgression of adaptive alleles and increase the potential interest of the study to readers of PRSB. Consequently, I am recommending that resubmission be allowed.

We addressed both reviewers' concerns by making substantial changes to the introgression analyses including new results, revising the interpretation and discussion of the introgression results, adding more details and analyses of the gut content and isotope data to bolster our support for the ecological intermediacy of this new species, and discussed some of the implications of the morphological differences in this species for its performance in the dietary niche of scale-eating. We now use a formal test of introgression and found similar support for candidate adaptive introgression regions.

Reviewer(s)' Comments to Author:

Referee: 1

Comments to the Author(s)

In this ms, Richards & Martin describe a new 'intermediate' wide-mouth scale-eater ecomorph in the pupfish adaptive radiation in the Bahamas using morphological and dietary analyses, and use whole genome resequencing data to investigate their evolution. They find that the new wide-mouth ecomorph is morphologically and trophically intermediate between the generalist and specialised scale-eater. Using genomic data, they find that the two scale-eater ecomorphs have diverged over 10kya, and that, contrary to expectations, that the shared 'scale-eater'-adaptive alleles swept earlier in the specialised scale-eater *D. desquamator* compared to the intermediate wide-mouth ecomorph. Comparisons of absolute divergence across the genome indicate that this pattern is potentially due to introgression of adaptive alleles between scale-eater ecomorphs.

Overall, this is a very interesting study, and I liked the approach of using this ecomorphologically intermediate ecomorph to test hypotheses on the evolution of the scale-eater

species and the underlying adaptive alleles. I have very much enjoyed reading this paper and only have a few comments (see below).

1. L. 185 – 191 (L. 219 – 227 in Suppl.): While the morphological differences are quite pronounced in the wild caught individuals, they seem to be quite a lot weaker in the lab-reared once. I was wondering if the differences are statistically significant, as 2 – 4% seem quite small and the confidence intervals overlap quite a bit for insertion height (Fig. S3A).

We agree that there is some phenotypic plasticity in these traits such that morphological differences between species are less pronounced among lab-reared individuals than observed in wild caught individuals. There doesn't appear to be a universal trend across all traits measured though and some traits appear more plastic than others, such as the preopercular insertion height noted by the reviewer in which the notable differences between wild caught individuals disappears in the lab-reared individuals.

We have updated figure S3 to compare morphological divergence among species in the wild and lab-reared populations and highlighted any overlapping 95% confidence intervals between species in morphological measurements for each of the 8 morphological traits measured.

From this new figure, we can see that some of the traits that are significantly distinct in the wild population of *C. sp.* 'wide-mouth' such as lower jaw length and buccal width remain significantly distinct in the lab-reared populations. The weaker divergence of preopercular insertion height among lab-reared individuals is interesting because preopercular insertion height is a proxy measurement for the cross-sectional attachment of the adductor mandibulae muscle. The plasticity observed in this musculature may be due to muscle plasticity associated with lack of scale-eating attacks in the lab. In the lab-reared colonies, all species were reared on the same diet, such that the scale-eating populations do not scale-feed as often as they would in the wild (but this still occurs with tankmates). Future studies will need to explore more thoroughly the potential developmental plasticity occurring in some of these traits observed here. We have also incorporated more discussion of the lab-reared morphological results into the main text.

Main text addition:

“Morphological traits were heritable in a common garden laboratory environment after one generation: lab-reared *C. sp.* 'wide-mouth' displayed significantly larger buccal width than *C. desquamator* (t-test; $P = 0.003$) and maintained their characteristic intermediate jaw lengths (ANOVA; $P = 0.03$, figure S3). There was also some evidence of phenotypic plasticity in both *C. desquamator* and *C. sp.* 'wide-mouth' compared to wild individuals likely caused by the common lab diet. See supplementary results for more details.” (Lines 199-204)

Supplemental text addition:

“Two of the focal three traits that distinguished wild populations of *C. sp.* 'wide-mouth' from the other pupfish species, lower jaw length and buccal width, remain significantly distinct in the lab-

reared populations (based on non-overlapping 95% confidence intervals around mean residuals of each species). However, one trait that distinguishes *C. sp.* ‘wide-mouth’ in the wild appears to be quite plastic under laboratory rearing conditions; there were no significant differences in mean adductor mandibulae height residuals among species in laboratory environments. This is interesting because adductor mandibulae height is a proxy measurement for the cross-sectional width of the adductor muscle directly beneath the skin. The plasticity observed in this proxy may be due to muscle plasticity. In the lab-reared colonies, all species are reared on the same diet, such that the scale-eating populations do not scale-feed as often as they would in the wild and thus lab-reared populations of scale-eaters may not develop the notably increased adductor muscle mass. Interestingly, the distinct differences in adductor mandibulae insertion height among wild populations disappeared in the lab-reared populations (figure S3), indicating phenotypic plasticity associated with the common diet of pellet foods and lack of scale-eating attacks in the lab.” (Lines 362-376)

2. Also, is there any indication of how biologically relevant such small morphological differences are in this system? I am not suggesting that they are not meaningful, I am more intrigued about any information, which would be interesting to include in the discussion (especially since morphological differences are not discussed in the main discussion).

We also find it intriguing that rather small changes in morphology in this system could be biologically relevant to the major ecological transitions observed here. Previous work in the lab suggests that the kinematics of scale-eating strikes, particularly that of peak gape distance, influences scale-feeding performance, such that there is an optimal distance for these species to open their jaws to take the biggest bite they can in a single strike (St. John et al. 2020 JEB). This peak gape size is in part controlled by the jaw length of a fish, where often a simple doubling in jaw length can result in a doubling in gape distance under simple geometric modeling of the jaw lengths and gape distance as an isosceles triangle (Ferry-Graham et al. 2010). We are currently measuring feeding kinematics trials of *C. sp.* ‘wide-mouth’ to assess the significance of their morphology on scale-feeding performance, but these results are beyond the scope of this manuscript. However, we added discussion of the biological relevance that small changes in some of these craniofacial morphologies have on feeding performance in these fish to the main text in the context of our prior research on scale-eating kinematic performance in this system.

Main text addition:

“Small modifications in craniofacial morphology among these species have major impacts on scale-eating performance in this system by altering kinematic traits such as peak gap size which is partially controlled by the length of the lower jaw, jaw protrusion distance, and the angle of the lower jaw relative to the suspensorium (St. John et al. 2020).” (Lines 190-193)

3. L. 259: Do you mean fig. 5E here?

Thank you for catching this. We have fixed typo to refer to the correct panel.

4. L. 287: You refer to the GO term panel here. I think there went something wrong with the figure panel order. But figure 5 and the legend are very nice and clear otherwise.

Thank you for catching this. We have changed the in-text figure panel reference to refer to the correct panel letter for the GO terms. The main text figures have been modified to reflect the change in focus for the manuscript and the full version of this figure with the GO terms is now in the supplement.

5. L. 289 – 290: How do you know that these SNPs are in regulatory regions of those genes? Were those SNPs annotated using a particular software, e.g. snpEFF or did you define regulatory regions based on the distance to the gene? Would be good to provide this information.

Thank you for the suggestion to clarify. We defined putative regulatory regions as those within 20-kb of a gene. We have clarified this in the main text and supplement.

6. L. 350 – 359: While low d_{xy} values are a good indicator for the introgression of these alleles from one specialist into the other, I was wondering if you have considered using more specialised approaches to test for the presence of introgression around shared adaptive loci compared to the remainder of the genome (e.g. RNDmin; Rosenzweig et al. 2016)? This would give the argument more weight and support the conclusions regarding the shared adaptive alleles a bit more.

RNDmin is a great suggestion for a formal test of introgression between sister species. Unfortunately, when attempting to use it on our dataset and reading the papers associated with the test statistic, we discovered that its accuracy and sensitivity depends heavily on the availability of haplotype information (Rosenzweig et al. 2016), which we do not have in our unphased genetic dataset. We did try running RNDmin and largely found that many regions in the genome were assigned a value of zero, which we believe are false positives (low values of RNDmin indicating introgression) caused by lack of haplotype information to calculate the lowest genetic difference between populations and individuals in our dataset. So alternatively, we used a similar statistic to the one the reviewer suggested: Relative Node Depth (RND; (Feder et al. 2005)), which doesn't rely on haplotype data but does offer more control for confounding factors than D_{xy} does as a measure of introgression. For example, it divides the absolute genetic distance between the two species of interest by the absolute genetic distance to an outgroup for each species to control for local variation in mutation rate.

We additionally ran no-gene-flow simulations of genetic divergence between populations of the same size and divergence time that we empirically estimated from the Osprey Lake populations to determine significance thresholds for detecting introgression using these statistics. We called regions that had an RND below the 5% percentile of the no-gene-flow simulated RND values as good candidate regions for introgression between the *C. sp.* 'wide-mouth' and *C. desquamator*. This changed our results slightly from the previous candidates based solely on D_{xy} , but all the previous candidates were further supported by this formal test of introgression. An additional

region with a shared adaptive allele was added to the list of candidate regions based on our no-gene-flow simulation thresholds. Additionally, we assessed whether any of these regions appeared as introgressed between the generalist *C. variegatus* and *C. desquamator* based on this RND statistic to control for signatures of current gene flow due to sympatry and incomplete reproductive isolation between all species in the same pond in general. We did not find any of these regions appeared to be introgressed with the generalist. We also calculated the RND values for the regions uniquely sweeping in *C. desquamator* and *C. sp* ‘wide-mouth’ to ensure that strong background selection in these regions relevant for scale-eating is not strongly biasing us to detect signatures of introgression in regions that contain a shared adaptive allele between the two scale-eating populations compared to those unique and distinct adaptive alleles between the two scale-eating populations. There is a much stronger signature of introgression detected using the RND statistic in regions of those shared adaptive alleles than those that have unique adaptive alleles (Table S10), supporting that strong selection against deleterious variation in these important regions is not the predominant cause of these adaptive introgression signatures. However, we acknowledge the point both reviewers made about how tricky it is to confidently tease apart confounding signatures from adaptive introgression and so we have also included more discussion of alternatives in the main text.

The following caveats about confounding factors has been included in the results section of the main text:

‘Alternatively, this genetic similarity may also be caused by strong background selection on shared ancestral variation. Effective population sizes are not drastically different between the two species and exon density is not in the upper tail of the genome wide-distribution (figure S1;table S10), two factors found in other studies where background selection tends to confound adaptive introgression inferences (Kim et al. 2018; Zhang et al. 2020). However, we do not have extensive knowledge of recombination breakpoints in this non-model system to distinguish between strong background selection on shared ancestral variation and adaptive introgression scenarios for each candidate adaptive introgression region.’ (Lines 381-389)

Referee: 2

Comments to the Author(s)

The manuscript by Richards et al. presents 24 genomes of a new weekly-specialized scale eater pupfish species that occurs sympatrically with three other species, including a strongly specialized scale eater to which the new wide-mouth species is sister. The manuscript combines a diversity of well-chosen analyses to generate a good understanding of the radiation and particularly of the wide-mouth species which had previously not been studied genomically.

Major points:

1. While I do agree that diversity can beget diversity, I do not see how this manuscript adds evidence to this hypothesis. Having this even in the title is thus not correct in my opinion. I would recommend strongly toning down this part.

We think these results are an exciting step in our efforts to investigate the various mechanisms underlying how major ecological transitions occur during adaptive radiation and whether genetic variation used in one species' adaptation could be utilized by species in the same radiation to promote further diversification. We also agree with the reviewer that the arguments of direct evidence for it from the results presented here can be toned down with the inclusion of caveats and more nuance and we followed the reviewer's suggestions. The helpful comments of the reviewer led to substantial changes in our analyses and interpretation of the results (please see our responses below labeled as Major Points 2-5 for specific changes made in analyses, focus and tone of manuscript). We have changed the title of this manuscript and kept only a brief mention in the discussion of how these results might relate to the general hypothesis of Whittaker 1977 that diversity can beget further diversity to the discussion. More generally, we have modified the introduction and discussion to include more focus on more specific mechanisms that appear involved in the major ecological transitions observed in this system, including more discussion what our new results on the origins of alleles and timing of selective sweeps say about the adaptive walks taken by both scale-eater species.

The new title of the manuscript is as follows:

“We get by with a little help from our friends: shared adaptive variation provides a bridge to novel ecological specialists during adaptive radiation”

The only mention of Whittaker's 1977 hypothesis about diversity begets diversity in relationship to our findings is in the discussion :

“An intriguing implication of these findings is that hybridization may allow different species to share many of the same adaptive alleles to occupy distinct but similar niches, in line with the syngameon and ‘diversity begets diversity’ hypotheses of adaptive radiation [18,43].” (Lines 395-398)

2. In the manuscript, the authors imply independent evolution of scale eating aided by introgression. However, the two scale eating species show up as sister species in all analyses. The finding of most sharing of «specialist alleles» between *desquamator* and *wide-mouth* is to be expected as these «specialist alleles» are inferred as alleles divergent from *variegatus*. As *desquamator* and *wide-mouth* are sister species, all alleles fixed in their common ancestor would be included here and would be shared by both scale eating species even if they are not involved in adaptation to scale eating.

We followed this suggestion and removed any implication that scale-eating evolved independently in the two scale-eater sister lineages. We agree that any conclusion about the origins of the *wide-mouth* species depends on inferences of ancestral states, which are extremely challenging or impossible to make. In this updated version of the manuscript, we have added formal tests of introgression and additional admixture analyses in response to reviewer 1's Major Point 6 and Reviewer 2's Major Points 2, 4 and 5. Our added analyses led to the discovery that *C. sp* 'wide-mouth' may have experienced introgression from a distant generalist outgroup lineage related to a previously sampled coastal North Carolina population. We address these new findings and add analyses in more detail in our response to reviewer 2's Major Point X below. While we think our added analysis strengthens our argument of a role of introgression in the

evolution of *C. sp.* ‘wide-mouth’, we recognize that we can’t say for certain that the adaptive alleles they share with *C. desquamator* are not from their shared common ancestor on San Salvador Island nor necessary for scale-eating adaptation unless we genetically manipulate them in the lab and we therefore add information about this possibility to the discussion. We have also balanced the focus of the manuscript and its conclusions between the shared and unique adaptive alleles so that the importance of the paper does not predominantly rely on evidence of introgression underlying the shared adaptive alleles between the two scale-eating species. These broader changes to balance the focus and interpretation of the results can be seen in the new discussion section of the manuscript on lines 360-444.

The following has been added to the discussion to give a caveat to the adaptive introgression between sister species finding:

‘Alternatively, this genetic similarity may also be caused by strong background selection on shared ancestral variation. Effective population sizes are not drastically different between the two species and exon density is not in the upper tail of the genome wide-distribution (figure S1;table S10), two factors found in other studies where background selection tends to confound adaptive introgression inferences (Kim et al. 2018; Zhang et al. 2020). However, we do not have extensive knowledge of recombination breakpoints in this non-model system to distinguish between strong background selection on shared ancestral variation and adaptive introgression scenarios for each candidate adaptive introgression region. ’ (Lines 381-389)

3. The fact that *C. brontotheriodes* had fewest alleles is also expected, given that it is very closely related to *variegatus*.

We have toned down the language implying that this was a surprising finding and make the shared selective sweeps across specialist results a more cohesive part of the paper by putting it into the context of genomic backgrounds constraining adaptive walks. We have put less emphasis on the comparison of adaptive alleles across all three specialists by moving a portion of the shared specialist allele findings to the supplementary results.

For example, we have reduced the description of these shared adaptive allele comparisons across all species in the results to:

‘Lastly, we compared the genetic divergence and selection patterns observed in the two scale-eating specialists to the divergent molluscivores specialist *C. brontotheriodes* to investigate the extent of allele sharing among all three trophic specialists in this adaptive radiation. We found that no fixed or nearly-fixed alleles relative to the generalist *C. variegatus* were shared across all three specialists (figure S9-S10; supplementary results). However, we did find evidence of 44 shared selective sweeps across all three specialist populations that were not shared with *C. variegatus* populations (figure S12C). These shared regions were significantly enriched for genes annotated for metabolic processes (figure S12D), suggesting shared selection for metabolizing the more protein-rich diet across all three trophic specialists (also see [39]). ’ (Lines 350-358)

Additionally, in the discussion we have added the following text to place the shared selective sweeps in context with the one of the new major focuses of the manuscript on what shared

selection on the same genetic variation implies about the microevolutionary processes underlying the major ecological transitions in this system.

‘We also found evidence for shared selective sweeps across all three specialists in regions that are enriched for genes annotated for metabolic processes such as short chain fatty acid and propionate metabolism (figure S12D). The lack of fixed alleles in these regions relevant to dietary specialization suggests polygenic selection (see supplemental for more discussion). Subtle shifts of allele frequencies across the genome can lead to divergent genomic backgrounds that give populations access to different ecological niches (e.g. [48,49]).

’ (Lines 419-424)

4. Similarly, the particularly low D_{xy} between the two scale-eating species at genomic regions with shared adaptive alleles does not imply introgression. It could just be a region of strong background selection or less gene flow with other species. As the two scale-eating species are sister species, no introgression is needed for allele sharing. The authors argue that this is unlikely because the age of the sweep is inferred to be different between the species. However, all those regions with low D_{xy} show the smallest difference in the selective sweep age, whereas the two regions with average or high D_{xy} show the largest difference in sweep age between the two species. The order of sweep age in the low D_{xy} regions is broadly consistent between the two species, indicating that they are correlated. I do not know the caveats of this method well enough to judge it, but I assume some demographic differences could contribute to the consistent shift in sweep age between the species. In almost all cases, the sweep age of shared alleles is older than that of unique alleles. This again suggests that the shared alleles could just be ancestral alleles, whereas the unique alleles are more recent alleles. None of the alleles that swept very recently are shared, speaking against recent introgression.

The reviewer makes two critiques in this response that we agree with and have addressed: 1) that low D_{xy} and shared selection signatures between two scale-eating populations could be caused by other processes than adaptive introgression such as background selection in one or both species and 2) that the differences in timing of selective sweeps between the two species provides weak evidence for adaptive introgression over shared selection in the common ancestor and could be used to argue against recent introgression instead. The specific new analyses and caveats we have added to the manuscript to address these comments are organized by each critique below:

1) Background selection’s confounding effects on adaptive introgression inferences:

Based on a suggestion made by reviewer 1 (Point 6), we have used a formal test of introgression instead of D_{xy} – Relative Node Depth (RND). RND is based on D_{xy} , but tries to control for the effects of mutation rate variation across the genome on introgression inferences. It does this by comparing the D_{xy} observed between focal species to the D_{xy} observed in an outgroup population, such that regions of low D_{xy} due to low mutation rates over several related lineages will not be flagged as a potential introgression candidate. Therefore, RND statistic results should filter out confounding signatures of low genetic dissimilarity due to conserved regions of low mutation rate, a signature often associated with strong background selection against deleterious variation.

However, we acknowledge that this statistic doesn't completely address the reviewers concern as this test assumes that mutation rate does not vary across the lineages used and we do not yet know whether that assumption holds for the groups we used in our tests. Its still possible that strong background selection private to the two scale-eating sister species could be confounding our signatures of adaptive introgression. Therefore, we have also added a new analysis looking for a pattern known from recent literature to cause false positive signatures of adaptive introgression from background selection on deleterious alleles: hybridization between populations with different effective population size and regions with high exon density (Kim et al. 2018; Zhang et al. 2020). We found that any given candidate adaptive introgression region in our study never exceeds the 55th percentile of the genome-wide distribution for exon density as calculated with 10-kb windows (shown in new table S10). Additionally, population sizes are very similar between *C. sp.* 'wide-mouth' and *C. desquamator* (Figure S1). However, we do not yet have a recombination map for any of these pupfish species, so we cannot do the extensive simulation testing recommended by the papers for best distinguishing adaptive introgression from background selection. Therefore, we have added the caveats to the main text, toned down several arguments for having strong evidence for adaptive introgression between sister species.

We added the new RND statistic results to the main text to replace the D_{xy} -based results from the previous version of the manuscript:

“Using the Relative Node Depth (RND) statistic, we also discovered that 5 of the 6 shared sets of adaptive alleles (all except for the unannotated region on scaffold 43;table S6) appear introgressed between *C. desquamator* and *C. sp.* 'wide-mouth', suggesting a substantial contribution of introgression to the adaptive alleles observed in *C. sp.* 'wide-mouth'.” (Lines 308-312)

But we have also included caveats about this result of adaptive introgression between sister species in the discussion section of the main text as mentioned in our response to reviewer 2's Main Point 2.

2) The weak support of selective sweep timing results for discerning between scenarios of adaptive introgression or selection in common ancestor

A) We agree with the reviewer that these signatures are complex and several processes could have resulted in this pattern, and the example they suggest is a valid one. We have removed the emphasis in manuscript on determining whether the shared adaptive alleles were selected upon in the ancestor, independently from segregating variation that was standing in both species after their initial divergence from each other, or introgressed from one species into the other by deleting the results and discussion strongly arguing for introgression as the correct scenario given our timing results. We have additionally added more nuance and caveats of interpretations throughout.

For example, in the discussion section we have added the following text to give alternative scenarios more weight for the timing of selective sweeps:

‘While *C. sp.* ‘wide-mouth’ is ecologically intermediate in its scale-eating behavior, our estimates of the relative timing of selective sweeps suggest that these shared alleles were first selected upon in the more specialized scale-eater *C. desquamator*, although unaccounted for demographic differences may also be contributing to differences in estimated timing between species.’ (Lines 373-377)

B) To address the second argument the reviewer makes here that sweep age of shared alleles is older than that of unique alleles which could be expected regardless of adaptive introgression, we have additionally expanded our assessment of the spatial source of candidate adaptive alleles unique to *C. sp.* wide-mouth and *C. desquamator* to match the previously included table with such information for the shared adaptive alleles. Intriguingly, none of the adaptive alleles unique to the *C. sp.* wide-mouth that we could estimate timing of selection on are de novo to San Salvador Island. These alleles all exist as standing genetic variation yet have more recent timing estimates than the shared adaptive alleles with *C. desquamator*, which includes a shared de novo allele to San Salvador Island. In opposition, the most recent sweeps for adaptive alleles unique to *C. desquamator* are predominantly from de novo alleles to San Salvador Island. However, some of the timing estimates of sweeps from de novo alleles do overlap in 95% HPDs of sweep ages of alleles that exist as standing genetic variation, so the recent timing of sweeps and differences in timing between adaptive alleles is not solely due to the expected pattern of younger ages for de novo mutations.

We have included more description of the timing of selective sweeps across spatial source of alleles throughout the results section in the main text. For example:

“All other introgressed adaptive alleles in *C. sp.* ‘wide-mouth’ swept more recently than the sweeps of the adaptive alleles they share with *C. desquamator*, including the shared de novo allele, and were not under selection in outgroup generalist populations.”(Lines 324-325)

“Intriguingly, all but one of the de novo adaptive alleles in *C. desquamator* swept at the same time in the recent past (figure 5).” (Lines 326-327)

As well as more discussion of the spatial sources of genetic variation and the timing of selection on alleles from different sources in to the discussion section of the main text. For example:

“The distinct timing of selection across different adaptive alleles in both *C. desquamator* and *C. sp.* ‘wide-mouth’ suggests that the ecological transition from generalist to novel scale-eating specialist involved such an adaptive walk in which selection on a beneficial allele was contingent on prior fixation of other adaptive alleles in each specialists’ genetic background (see supplemental materials for further discussion). This is best highlighted by the pattern observed in *C. desquamator* in which nearly all de novo mutations

swept at the same time in a distinct selective stage from other adaptive variants rather than being selected upon as they originated (figure 5).” (Lines 408-411)

5. The authors have previously shown that introgression from San Salvador Island facilitated the evolution of the specialized scale eater. It would be good to better link this manuscript to previous work by showing if the unique alleles and the shared alleles are also found in the San Salvador Island population. In Table S7, the authors show this for the shared alleles. However, I would also like to see it for the unique alleles. Are the unique specialist alleles newly arisen or do they also confer old alleles that perhaps introgressed from other geographic regions?

We followed this suggestion and added new results and two additional tables to the supplement (table S7 and S8) that show the source of genetic variation for each variant that is uniquely fixed or nearly-fixed ($F_{st} > 0.95$) and under selection in the *C. sp* ‘width-mouth’. Briefly, we found that de novo alleles to San Salvador Island were predominantly swept to near-fixation in *C. desquamator* (coded red: figure 5). *C. sp* wide-mouth population contained only 2 unique alleles that were de novo to SSI but seven of the unique to *C. wide-mouth* alleles fell inside candidate introgression regions between *C. sp* wide-mouth and a North Carolina population of *C. variegatus*. These new findings have been incorporated into the manuscript as new analyses described in the methods, new results, and new discussion in the main text. The modifications we made in response to the reviewer’s suggestion is organized below in three sections: new methods, new results, and new discussion

1. New methods and analyses

- A. We have added new analyses describing how we categorized alleles as de novo, introgressed or standing genetic variation following the protocol we used previously in (Richards et al. 2021). Alleles were determine as introgressed if they fell in a candidate adaptive introgression region between *C. sp* ‘wide-mouth’ and one of 5 outgroup generalist populations we sequenced in the previous study mentioned by the review. The main text and supplementary methods have been updated to include these new analyses.

The following has been added to the method section in the main text:

‘Lastly, we investigated the spatial origins of adaptive alleles shared and unique to the two scale-eating specialists by searching in our previous study spanning Caribbean-wide outgroup populations for these alleles [25]. Adaptive alleles were assigned as standing genetic variation if observed in any population outside SSI or de novo if they were only observed within populations on SSI. Additionally, we investigated signatures of introgression across the genome of *C. sp* ‘wide-mouth’ and *C. desquamator* to determine if they showed evidence of adaptive introgression from outgroup generalist populations as observed previously [25]. See supplementary methods for more details on introgression analyses.’ (Lines 174-181)

- B. We added additional genome-wide tests of differential introgression (f_4 -statistic) and admixture (qpgraph) involving distantly related generalist lineages and detected

signatures of secondary contact events and introgression between a North Carolina population of *C. variegatus* and *C. sp* ‘wide-mouth’. Description of these complementary analyses can be found in the methods section of the supplementary materials on lines 261-352.

2. New main text results, tables, and figures

We have added new results that highlight the spatial source of genetic variation for each candidate adaptive allele and discuss their implication on the evolutionary scenario underlying shared ecological niches between *C. wide-mouth* and *C. sp* wide-mouth. This new finding of introgression with a distant generalist lineage and the more robust tests of introgression between *C. desquamator* and *C. sp*. ‘wide-mouth’.

In the results section we have added the following paragraph:

“The adaptive alleles shared by *C. desquamator* and *C. sp*. ‘wide-mouth’ occurred as low frequency standing genetic variation in the Caribbean, with the exception of a single de novo allele on SSI located in an unannotated region on scaffold 6 (figure 5). The adaptive alleles unique to *C. desquamator* and *C. sp* ‘wide-mouth’ also predominantly came from standing genetic variation (84% and 81%, respectively). 14% of adaptive alleles unique to *C. desquamator* were de novo mutations to SSI and 2% occurred in candidate introgression regions (table S7). We found the opposite in *C. sp* ‘wide-mouth’: only 4% of their unique adaptive alleles were de novo mutations whereas 15% occurred in candidate introgression regions (table S8). This adaptive introgression was detected for generalist populations sampled from North Carolina and Laguna Bavaro in the Dominican Republic (table S8; figure S11).” (Lines 299-308)

3. New interpretations and discussion points

This new finding of introgression with a distant generalist lineage of *C. variegatus* from a North Carolina population and *C. sp*. ‘wide-mouth’ led us to highlight and discuss those adaptive alleles that are unique to each scale-eating species where previously we focused mainly on those that are shared. We now more extensively discuss how these patterns of some shared and many unique adaptive alleles, alongside the staggered timing of selection, indicate a strong role for epistasis. We discuss the powerful potential of epistatic interactions to not only play a role in multiple species making similar ecological transitions through shared genetic variation (be it introgressed or ancestrally shared) but also result in variable evolutionary outcomes even from the same starting place (i.e. occupying unique version of new ecological niches). We discuss the use of slightly different sources of genetic variation (de novo for *C. desquamator* and introgression for *C. sp* ‘wide-mouth’) in this new context as well.

We hope the new results and discussion contributes to a more interesting and balanced manuscript that doesn’t solely rely on having strong support for shared adaptive alleles

between sister species coming from introgression rather than selection upon in the ancestor. The details of all changes made throughout the manuscript to reflect these new results can be seen in the track changes present in new version, but we have included some highlights of the update main text below.

The discussion has been updated with new topics and results highlighted, including the following:

“We also found strong signatures of introgression in *C. sp.* ‘wide-mouth’ genomes from outgroup generalist populations that were not shared with *C. desquamator* (figure S11; table S11). Craniofacial morphology is a major axis of diversification between trophic specialists in this system [44], yet *C. sp.* ‘wide-mouth’ appears to have little unique genetic variation relevant for craniofacial traits compared to the other two specialists (figure S10). Despite this, they do exhibit transgressive craniofacial phenotypes not seen in the other specialists. An intriguing implication of these findings is that hybridization may allow different species to share many of the same adaptive alleles to occupy distinct but similar niches, in line with the syngameon and ‘diversity begets diversity’ hypotheses of adaptive radiation [18,40].” (Lines 390-398)

“This highlights the dual influence of epistatic interactions on adaptive walks in rugged landscapes – epistasis reduces number of available paths but increases the number of local fitness peaks populations can get stuck on [50]. Selection on the same adaptive alleles may have allowed both scale-eating species access to the same area of the fitness landscape but epistatic interactions with private mutations and introgressed variation in each lineage may have resulted in divergent paths to scale-eating, ultimately contributing to diverse evolutionary outcomes even from a shared starting point.

The use of adaptive alleles from distinct spatial sources, the distinct morphologies and divergent genomic backgrounds, and potential introgression of adaptive alleles from the more specialized scale-eater *C. desquamator* into *C. sp.* ‘wide-mouth’ reveals a tangled path for novel ecological transitions in nature. The complex epistatic interactions at microevolutionary scales implicated in this study make it all the more fascinating that novel ecological transitions are a common macroevolutionary feature of adaptive radiation. “ (Lines 432-444)

6. Given that only half of the *C. sp.* ‘wide-mouth’ individuals contained scales in the stomach content. What was the rest of their diet? This would be very important to know.

We have updated the main text with more discussion of what we know about the diet of *C. sp.* ‘wide-mouth’ in comparison to the diets of the other species in the radiation and incorporated new analyses of the dietary data. We have placed *C. sp.* ‘wide-mouth’ diet findings into better context with the other species and ecology of this radiation by 1) including more discussion of

the other gut contents discovered beyond scales and across species in different lakes and 2) by adding new statistical analyses of isotope data commonly used in other studies to assess ecological overlap between populations. The specific changes made are organized separately for these two points below.

1) Expanding on the gut contents of *C. sp. 'wide-mouth'* to place them in more ecological

The following has been added to the main text results:

“Detritus made up the rest of the *C. sp. 'wide-mouth'* and *C. desquamator* diets and was the dominant component of *C. variegatus* gut contents, except for a single individual with one mollusc shell. A previous study that characterized contents of *C. variegatus*, *C. brontotheroides*, and *C. desquamator* populations across several ponds also found detritus to be the dominant component of each species' diet (49-71%) and nearly zero scales in the gut contents of *C. variegatus* and *C. brontotheroides* [33].” (Lines 220-225)

The following has been added to the supplemental text:

“ We found that *C. sp. 'wide-mouth'* ingest scales, but at a significantly lower frequency than *C. desquamator* (Wilcoxon Rank Sum test, $P = 0.004$; figure 3A). We did not detect any scales in *C. variegatus* guts (figure 3A). Detritus made up the only other gut contents present besides scales in all the specimen of *C. sp. 'wide-mouth'* from Osprey Lake that we dissected. This was similar for the other scale-eater *C. desquamator* individuals from Osprey Lake, in which we also found only scales and detritus in their gut. For the generalist species *C. variegatus*, their guts contained only detritus except for a single individual in which we found a snail shell. The low diversity in gut contents among species may be due to the relatively small sample size of individuals we had that we could perform gut content analysis on for this study ($n=10$ for each species). However, the predominance of detritus in the gut contents of Osprey Lake populations of all three species is not surprising given the thick ($<1m$) bottom layer of mud and flocculent in this lake(Edwards 2001). Additionally, detritus made the largest percentage (49-71%) of San Salvador Island pupfish diet in other lakes across the island as well. A previous study that conducted gut content analyses with 2-4X larger sample sizes also found that the majority of *C. variegatus*, *C. brontotheroides*, and *C. desquamator*'s diet is detritus. Except for one population of *C. desquamator* in Crescent Pond where scales made up 50% of their gut contents and detritus only made up 30%. Therefore, the ecological intermediacy of *C. sp. 'wide-mouth'* is supported by their intermediate ratio of scales to detritus, which is the predominant axis of dietary divergence between *C. variegatus* and *C. desquamator* in this lake. The lack of other dietary items in the gut contents of *C. sp. 'wide-mouth'* further supports our arguments that their ecological divergence is along the same specialist axis of scale-eater *C. desquamator* and that they share little ecological overlap with the other molluscivore specialist *C. brontotheroides*.” (Lines 377-392)

2) Exploring evidence for intermediate ecology based on long-term dietary niche

Additionally the isotope data on an independent set of individuals provides insights into how this intermediate level of scale-eating might play into the realized trophic level of *C. sp* wide-mouth relative to *C. variegatus* and *C. desquamator*. Even though *C. sp* 'wide-mouth' and *C. desquamator* have a large overlap with *C. variegatus* in the detritus portion of their diets, we do still all three species occupying different areas of niche space in the form of different trophic levels. Previous characterization of $\delta^{15}\text{N}$ and $\delta^{13}\text{C}$ isotopes in muscle tissue from *C. variegatus*, *C. brontotheroides* and *C. desquamator* show that *C. desquamator* has the highest $\delta^{15}\text{N}$ values on average (Martin and Wainwright 2013), which is an indication that they occupy a higher trophic level than the other species, particularly the generalist *C. variegatus*. In our reanalysis with isotope data from Osprey Pond populations that now includes isotopes for *C. sp* 'wide-mouth', we see complementary support for that intermediate ecology observed in the gut content analysis, where *C. sp*. 'wide-mouth' occupies a distinct and intermediate trophic level between *C. variegatus* and *C. desquamator*. We have added more statistical analysis of the isotope data to help support that this new population resides in a distinct and intermediate trophic niche between *C. variegatus* and *C. desquamator* with ANOVA and effect size characterization of the differences in $\delta^{15}\text{N}$ levels between all three populations (Fig S4; supplementary results). We have also calculated common measures of niche overlap such as standard ellipse area statistics and bivariate means and confidence intervals using the R package SIBER to visualize the amount of ecological overlap between species on our isotope biplots. Even with the caveat that these overlap measurements rely on $\delta^{13}\text{C}$ values, and we don't yet know how similar the sources of carbon in each species diet are (different sources of carbon can have similar $\delta^{13}\text{C}$ values creating a false signature of high dietary overlap along this axis), we can see each species is fairly distinct in trophic level from each other based on the lack of overlap in 95% confidence intervals in the bivariate means for each population and the standard ellipse area corrected for sample size (figure 3B).

The following has been added to the main text alongside modifications to panel B in figure 3:

“Dietary overlap was characterized by comparison of population mean scale count from gut contents using ANOVA and ellipse areas and bivariate means on isotope biplots using SIBER [24]. See supplemental methods for more details on sample sizes and analyses.” (Lines 122-124)

“The intermediate scale-eating dietary niche of the wide-mouth ecomorph is complemented by our stable isotope analyses, which provide long-term snapshots of the carbon sources and relative trophic levels in these species. Osprey Lake individuals collected on the same day from the same site differed in $\delta^{15}\text{N}$ levels across species (ANOVA, $P = 4.55 \times 10^{-6}$; figure 3B and S4); 'wide-mouth' $\delta^{15}\text{N}$ was intermediate between *C. variegatus* and *C. desquamator* (Tukey HSD; $P = 1.34 \times 10^{-5}$ & 1.11×10^{-4} respectively), supporting its intermediate trophic position. SIBER analyses of trophic niche position indicate distinct positioning of the wide-mouth ecomorph based on the lack of extensive overlap in niche space measured by standard ellipse areas and bivariate

means with 95% confidence intervals of isotope values among the species (figure 3B).” (Lines 226-235)

The following description of methods for analyzing overlap among species from the isotope data has been added to the methods section of the supplemental text:

“ $\delta^{13}\text{C}$ and $\delta^{15}\text{N}$ levels were summarized across each of the three species to look at overlap among species by 40% and 95% Standard Area Ellipses and 95% confidence ellipses around bivariate means on isotope biplots (figure 3) of the data using R package SIBER (Jackson et al. 2011). Additionally we assessed whether there were significant differences in $\delta^{15}\text{N}$ values between species through an ANOVA using `aov()` function from the base statistics package in R (v 4.1.0).” (lines 72-76)

Minor points:

1. L. 58: Unclear what stages means here. The two cited papers do not talk about stages...

We added additional information about what these stages refer to in this introduction sentence, clarifying that the stages referred to in these references come from a framework of adaptation occurring in stages of potentiation, actualization, and refinement both genetically and morphologically.

“Recent conceptual frameworks for understanding adaptation to novel fitness peaks suggest that these major ecological transitions likely occur in stages of potentiation, actualization and refinement (Blount et al. 2012; Erwin 2021).” (Lines 59-61)

2. Why is the molluscivore (*C. brontotheriodes*) not included in the phenotypical and stomach content analysis? This seems relevant.

We have clarified the exclusion of molluscivores for these analyses in the main and supplementary text. The molluscivores were not included in the morphological and gut content analyses because we were specifically interested in characterizing the traits associated with adaptation to the ecological continuum between generalist and scale-eating niches. Previous extensive characterization of the molluscivores *C. brontotheriodes* to the generalist *C. variegatus* and scale-eater *C. desquamator* show that molluscivores have opposing phenotypes to this scale (e.g. even shorter oral jaws lengths than the generalists) and same low level of scales in their gut contents to the generalist as well but additional dietary diversity in the form of ostracods and gastropods making up ~22-30% of their gut contents. We also did not find any gastropod or ostracod remnants in the gut contents of *C. sp* wide-mouth and thought it would therefore be adding irrelevant information to the ecological divergence along the scale-eating specialization axis. We have clarified in the main and supplementary text why we think molluscivores represent a largely distinct specialist axis in this system in terms of morphology, ecology and genetics and include more written reference where comparisons that include the molluscivores have been made in other studies to support this conclusion.

The following description of *C. sp* ‘wide-mouth’ morphological traits compared to *C. brontotheroides* has been added to the main text:

“*C. sp* ‘wide-mouth’ also did not show morphological divergence comparable to that observed in the molluscivore *C. brontotheroides*. The molluscivore specialist presents an opposing axis of morphological divergence to the scale-eating specialists, with shorter oral jaw length and larger eye diameters than even the generalist *C. variegatus*, in addition to a novel nasal protrusion of the maxilla not observed in any other Cyprinodontidae species (Martin and Wainwright 2013).” (Lines 194-198)

The following description of *C. sp* ‘wide-mouth’ ecology and diet compared to *C. brontotheroides* has been added to the main text:

“A previous study that characterized contents of *C. variegatus*, *C. brontotheroides*, and *C. desquamator* populations across several ponds also found detritus to be the dominant component of each species’ diet (49-71%) and nearly zero scales in the gut contents of *C. variegatus* and *C. brontotheroides* [33].” (Lines 222-225)

3. Table S1: Including an admixture test testing for a topology that is inconsistent with the genome-wide average tree and the demographic models, is more confusing than helpful in my opinion. I would remove the bold lines.

We have deleted these tests from table S1 as the reviewer suggested.

Fig. 6 is beautiful and a very nice summary of all the ages.

Thank you for the compliment, we really like this figure too! We modified this figure by adding information about the spatial source of genetic variation, but largely left it as is.

References

- Blount, Z. D., J. E. Barrick, C. J. Davidson, and R. E. Lenski. 2012. Genomic analysis of a key innovation in an experimental *Escherichia coli* population. *Nature* 489:513–518.
- Edwards, C. D. 2001. Effect of Salinity on the Ecology of Molluscs in the Inland Saline Waters of San Salvador Island: A Experiment in Progress.
- Erwin, D. H. 2021. A conceptual framework of evolutionary novelty and innovation. *Biol. Rev.* 96:1–15.
- Feder, J. L., X. Xie, J. Rull, S. Velez, A. Forbes, B. Leung, H. Dambroski, K. E. Filchak, and M. Aluja. 2005. Mayr, Dobzhansky, and Bush and the complexities of sympatric speciation in *Rhagoletis*. *Proc. Natl. Acad. Sci. U. S. A.* 102 Suppl:6573–80.
- Ferry-Graham, L. A., L. P. Hernandez, A. C. Gibb, and C. Pace. 2010. Unusual kinematics and jaw morphology associated with piscivory in the poeciliid, *Belonesox belizanus*. *Zoology* 113:140–147.

- Fragata, I., A. Blanckaert, M. A. Dias Louro, D. A. Liberles, and C. Bank. 2019. Evolution in the light of fitness landscape theory. *Trends Ecol. Evol.* 34:69–82.
- Hansen, T. F. 2013. Why epistasis is important for selection and adaptation. *Evolution* (N. Y). 67:3501–3511.
- Jackson, A. L., R. Inger, A. C. Parnell, and S. Bearhop. 2011. Comparing isotopic niche widths among and within communities: SIBER - Stable Isotope Bayesian Ellipses in R. *J. Anim. Ecol.* 80:595–602.
- Kim, B. Y., C. D. Huber, and K. E. Lohmueller. 2018. Deleterious variation shapes the genomic landscape of introgression. *PLoS Genet.* 14:1–30.
- Martin, C. H., and P. C. Wainwright. 2013. On the measurement of ecological novelty: scale-eating pupfish are separated by 168 my from other scale-eating fishes. *PLoS One* 8:e71164.
- Martin, C. H., and P. C. Wainwright. 2011. Trophic novelty is linked to exceptional rates of morphological diversification in two adaptive radiations of cyprinodon pupfish. *Evolution* (N. Y). 65:2197–2212.
- McGirr, J. A., and C. H. Martin. 2018. Parallel evolution of gene expression between trophic specialists despite divergent genotypes and morphologies. *Evol. Lett.* 2:62–75.
- Otte, K. A., V. Nolte, F. Mallard, and C. Schlötterer. 2021. The genetic architecture of temperature adaptation is shaped by population ancestry and not by selection regime. *Genome Biol.* 22:1–25. *Genome Biology*.
- Patton, A. H., E. J. Richards, K. J. Gould, L. K. Buie, and H. Christopher. 2021. Adaptive introgression and de novo mutations increase access to novel fitness peaks on the fitness landscape during a vertebrate adaptive radiation.
- Richards, E. J., J. A. McGirr, J. R. Wang, M. E. St. John, J. W. Poelstra, M. J. Solano, D. C. O’Connell, B. J. Turner, and C. H. Martin. 2021. A vertebrate adaptive radiation is assembled from an ancient and disjunct spatiotemporal landscape. *Proc. Natl. Acad. Sci. U. S. A.* 118.
- Rosenzweig, B. K., J. B. Pease, N. J. Besansky, and M. W. Hahn. 2016. Powerful methods for detecting introgressed regions from population genomic data. *Mol. Ecol.* 25:2387–2397. Wiley/Blackwell (10.1111).
- Seehausen, O. 2004. Hybridization and adaptive radiation. *Trends Ecol. Evol.* 19:198–207.
- St. John, M. E., R. Holzman, and C. H. Martin. 2020. Rapid adaptive evolution of scale-eating kinematics to a novel ecological niche. *J. Exp. Biol.* 223:jeb217570.
- Whittaker, R. 1977. Evolution of species diversity in land communities. *Evol. Biol.*
- Zhang, X., B. Kim, K. E. Lohmueller, and E. Huerta-Sánchez. 2020. The impact of recessive deleterious variation on signals of adaptive introgression in human populations. *Genetics* 215:799–812.